# SIFa peptidergic neurons orchestrate the internal states and energy balance of male *Drosophila melanogaster*

Yutong Song[1☯], Tianmu Zhang[1☯], Tae Hoon Ryu[2,3], Kyle Wong[4], Zekun Wu[1], Yanan Wei[1], Justine Schweizer[4], Khoi-Nguyen Ha Nguyen[4], Alex Kwan[4], Xiaoli Zhang[1], Kweon Yu[2,3], Woo Jae Kim [1,4]*

**1** The HIT Center for Life Sciences, Harbin Institute of Technology, Harbin, China, **2** Metabolism and Neurophysiology Research Group, Korea Research Institute of Bioscience and Biotechnology (KRIBB), Daejeon, Korea, **3** Department of Functional Genomics, University of Science and Technology (UST), Daejeon, Korea, **4** Department of Cellular and Molecular Medicine, University of Ottawa, Ottawa, Canada

☯ These authors contributed equally to this work.
* wkim@hit.edu.cn

## Abstract

Neuropeptide SIFamide (SIFa) neurons in *Drosophila melanogaster* have been characterized by their exceptionally elaborate arborization patterns, which extend from the brain into the ventral nerve cord (VNC). SIFa neurons are equipped to receive signals that integrate both internal physiological cues and external environmental stimuli. These signals enable the neurons to regulate energy balance, sleep patterns, metabolic status, and circadian timing. These peptidergic neurons are instrumental in orchestrating the animal's internal states and refining its behavioral responses, yet the precise molecular underpinnings of this process remain elusive. Here, we demonstrate that SIFa neurons coordinate a range of behavioral responses by selectively integrating inputs and outputs in a context-dependent manner. These neurons engage in a feedback loop with sNPF neurons in the VNC, modifying behaviors such as longer mating duration (LMD) and shorter mating duration (SMD). Additionally, SIFa neurons interact with dopamine and glutamate to differentially regulate sleep and mating duration. Activating SIFa neurons leads to reduced mating duration and increased food intake, while deactivating them reduces food intake. Overall, these findings demonstrate the importance of SIFa neurons in absorbing inputs and turning them into behavioral outputs, shedding light on animal's intricate behavioral orchestration.

## Introduction

The brain's activity is always changing, jumping from one state to another. This has a big impact on not only our neural responses to sensory inputs but also on our ability to process this huge amount of information, make decisions, and

Biology

and reproduction in any medium, provided the
original author and source are credited.

**Data availability statement:** All relevant
data are within the paper and its Supporting
information files.

**Funding:** This research was supported a
University of Ottawa Startup grant 602496
to WJK, Startup funds from HIT Center for
Life Science to WJK, a University of Ottawa
Interdisciplinary Research Group Funding
Opportunity (IRGFO stream 1 and 2) grants
148101 and 148747 to WJK, a Natural Sciences
and Engineering Research Council of Canada
(NSERC) Discovery grant (reference: 211406)
to WJK, a University of Ottawa Brain and Mind
Research Institute/Center for Neural Dynamics
Open call project grant 150950 to WJK, a
Mitacs Globalink Research Internship Program
grant 17268 to WJK. This research was also
supported by the Brain Pool Program of the
National Research Foundation in Korea, grant
ZYM5041911 to WJK, Burroughs Welcome
Fund Collaborative Research Travel Grants
(reference: 1017486) to WJK, and a NVIDIA
Academic Hardware Grant Program to WJK.
The funders had no role in study design, data
collection and analysis, decision to publish, or
preparation of the manuscript. SGL received
salary from the 'University of Ottawa Startup
grant to WJK' and HM from the 'Startup funds
from HIT Center for Life Science to WJK'.

**Competing interests:** The authors have
declared that no competing interests exist.

**Abbreviations:** 5-HT, serotonin; AG, abdom-
inal ganglion; AHL, artificial hemolymph;
AL,antennal lobes; ATP, adenosine tri-
phosphate; CB, central brain; CNS, central
nervous system; DA, dopamine; EX-Q, excreta
quantification; GRAB, GPCR-activation-based;
KLH, Keyhole Limpet Hemocyanin; LAL, lateral
accessory lobe; LMD, longer mating duration;
MD, mating duration; NPRs, neuropeptide
receptors; NPs, neuropeptides; NT, neurotrans-
mitter; OA,octopamine; PI, pars intercerebralis;
ROI, region of interest; SIFa, Neuropeptide
SIFamide; SIFaR, SIFa receptor; SMD, shorter
mating duration; sNPF, short neuropeptide F;
SPR, sex peptide receptor; VNC, ventral nerve
cord.

behave in the proper way [1,2]. Our decisions are frequently accompanied by 'internal states (or π states)' like motivation, arousal, desire, or emotion. Internal states have common characteristics, including persistence and scalability [3,4]. However, very little is known about how such internal states are recorded in animal brains and whether they impact behavioral decision-making directly or indirectly.

Internal states in reproductive social behaviors, such as mating and fighting, have been intensively studied [3,5–7]. Genetic methods in flies have enabled for the discovery of discrete neuronal groups expressing dopamine (DA) [8], serotonin (5-HT) [9], or octopamine (OA) [10], all of which control aggressive behavior. The male courtship circuitry of the fruit fly is one of the most intensively researched neural systems for understanding how the brain governs an intrinsic reproductive social behavior [6,7,11]. Nutrient homeostasis also creates unique internal states of the brain, which influence an animal's food preference [12,13]. The shift between internal states and their influence on behavioral decision-making has been studied in courtship, aggressiveness, and nutrient-specific appetites, but how these shifts are encoded and combined to modify specific choice is unclear [3,12–18].

Interval timing is a specific timescale that is measured precisely within the range of seconds to minutes [19]. Humans and other animals use interval timing for foraging, multi-step arithmetic, and decision-making [19–25]. Recent research suggests the brain encodes time as temporal changes in network states [26]. Observations into the interplay between sensory inputs and the impact of internal states like attention have been uncovered by psychophysical measures of interval timing [27].

The mating duration (MD) of male fruit flies is a suitable model for studying interval timing, and it could change based on internal states and environmental context. Previous studies by our group [28–30] and others [31,32] have established several frameworks for investigating the MD using sophisticated genetic techniques that can analyze and uncover the neural circuits' principles governing interval timing. In particular, males exhibit longer mating duration (LMD) behavior when they are exposed to an environment with rivals, which means they prolong their MD. Conversely, they display shorter mating duration (SMD) behavior when they are in a sexually saturated condition, meaning they reduce their MD [33,34].

Multiple systems provide evidence that neuromodulators, such as biogenic amines or neuropeptides, are crucial internal state and behavior regulators [35–38]. Internal states prolong external influences on behavior [39] and neuropeptides (NPs) may facilitate internal state perpetuation [3]. SIFa is a conserved neuropeptide (NP) in insects, crustaceans, and arachnids. This neuropeptide is expressed in four cells near pars intercerebralis (PI) and the neuronal process of these neurons extend throughout the neural system [30,40–43]. Studies have linked SIFa to hunger and feeding [44,45], courtship [41,42], sleep [40,46,47], and memory [48]. Here, we provide clear evidence that broadly arborized SIFa neurons record the internal states for two distinct interval timing behaviors.

## Results

### The expression of SIFa in neurons controls two unique interval timing behaviors in male *Drosophila melanogaster,* regardless of sexual dimorphism

To determine which NP is neuronally connected to both LMD and SMD (Fig 1A and 1B), we performed screening with pan-neuronal *elav^c155* driver with NPs and neuropeptide receptors (NPRs)-RNAi lines (S1–S5 Files, S1 Table). We discovered that knocking down the neuronal expression of CCHa1, CCHa2, ETH, ilp4, and SIFa eliminates both LMD and SMD behaviors (S1 Table and Fig 1C and 1D). SIFa is selected from the NPs that affect both LMD and SMD because it is known to govern internal state-related activities such as sleep, hunger, and mating [40,42,44–46]. The expression pattern of SIFa in four neurons located in PI is validated by utilizing neuronal-expressing *GAL80* line (S1A Fig), as well as anti-repo or anti-elav antibodies (S1B Fig). Using the *GAL4^SIFa.PT* driver and the *elav^c155* driver, we observed a significant decrease in SIFa immunoreactivity following *SIFa-RNAi* treatment, thereby confirming the effective knockdown of SIFa in these cells. In contrast, when *SIFa-RNAi* was expressed under the control of the *repo-GAL4* driver, no significant change in SIFa immunoreactivity was detected (S1C Fig). Notably, our custom-generated SIFa polyclonal antiserum, while exhibiting weaker overall signal intensity compared to previously reported antibodies [42], specifically labels SIFa neuronal cell bodies and is validated by SIFa-GAL4-mediated knockdown. This control experiment highlights the specificity of the SIFa-RNAi effect and supports the conclusion that the behavioral changes observed in SMD and LMD are likely attributable to the targeted reduction of SIFa in the intended neuronal populations.

Localized amplification of SIFa peptide level through the utilization of *UAS-SIFa* with *GAL4^SIFa.PT* influences both LMD and SMD behavior through either secretion (Fig 1E and 1F) or membrane-tethered formats (Fig 1G and 1H), suggesting that the level of SIFa peptide in SIFa neurons is important for controlling interval timing behaviors. In line with the effects of SIFa overexpression on LMD and SMD behaviors, both social isolation and sexual experience lead to an increase in SIFa transcript levels (S1D Fig). However, the SIFa peptide level remained consistent in the PI region despite variations in the social environment (S1F–S1H Fig). These findings with genetic control experiments (Fig 1I–1P) indicate that the neuronal function of the SIFa neuropeptide itself, plays a critical role in regulating interval timing behaviors.

In order to compare the distinctions between male and female SIFa-positive neurons in the central nervous system (CNS), we examined the distribution of SIFa-expressing cells in terms of their membrane and nuclear expression patterns, utilizing *UAS-CD4-tdGFP* and *UAS-RedStinger*. As previously described [43,44,42], a membrane marker expressing *GAL4^SIFa.PT* exhibits a broad arborization pattern in both the male and female central brain (CB) (Fig 1Q and 1R). We found no significant differences between male and female brain regions covered by the SIFa cell membrane (S1I and S1J Fig). While qualitative observations suggest more intense SIFa immunoreactivity in males and increased arborization patterns in female antennal lobes (AL), quantitative analysis showed no significant differences in regional coverage (S1I and S1J Fig). Precise morphological investigation would be necessary to discern potential sexual dimorphic differences in SIFa neuronal architecture. SIFa acts on *fruitless*-positive neurons via the SIFa receptor (SIFaR) and it has been reported that SIFa neurons do not express *fruitless*, which are responsible for the generation of male-specific neuronal circuitry [41]. Building on this foundation, we sought to determine if the sexual dimorphism in SIFa neurons contributes to the regulation of interval timing behaviors. We have validated that *GAL4^SIFa.PT* neurons do not label *fru^FLP*-positive neurons in the male brain (S1L Fig). Interval timing behaviors are unaffected by the expression of female form of *doublesex,* which drives female-specific splicing of downstream genes (*UAS-dsx^F*) (S2A and S2B Fig); *transformer*, a key player in female sex determination affecting *dsx* splicing (*UAS-tra^F*) (S2C and S2D Fig); or *sex lethal* (*UAS-sxl*) (S2E and S2F Fig) all of which have strong feminizing activity. According to these results and genetic control experiments (Fig 1I–1P), neurons that express SIFa are not implicated in sexual dimorphism by itself.

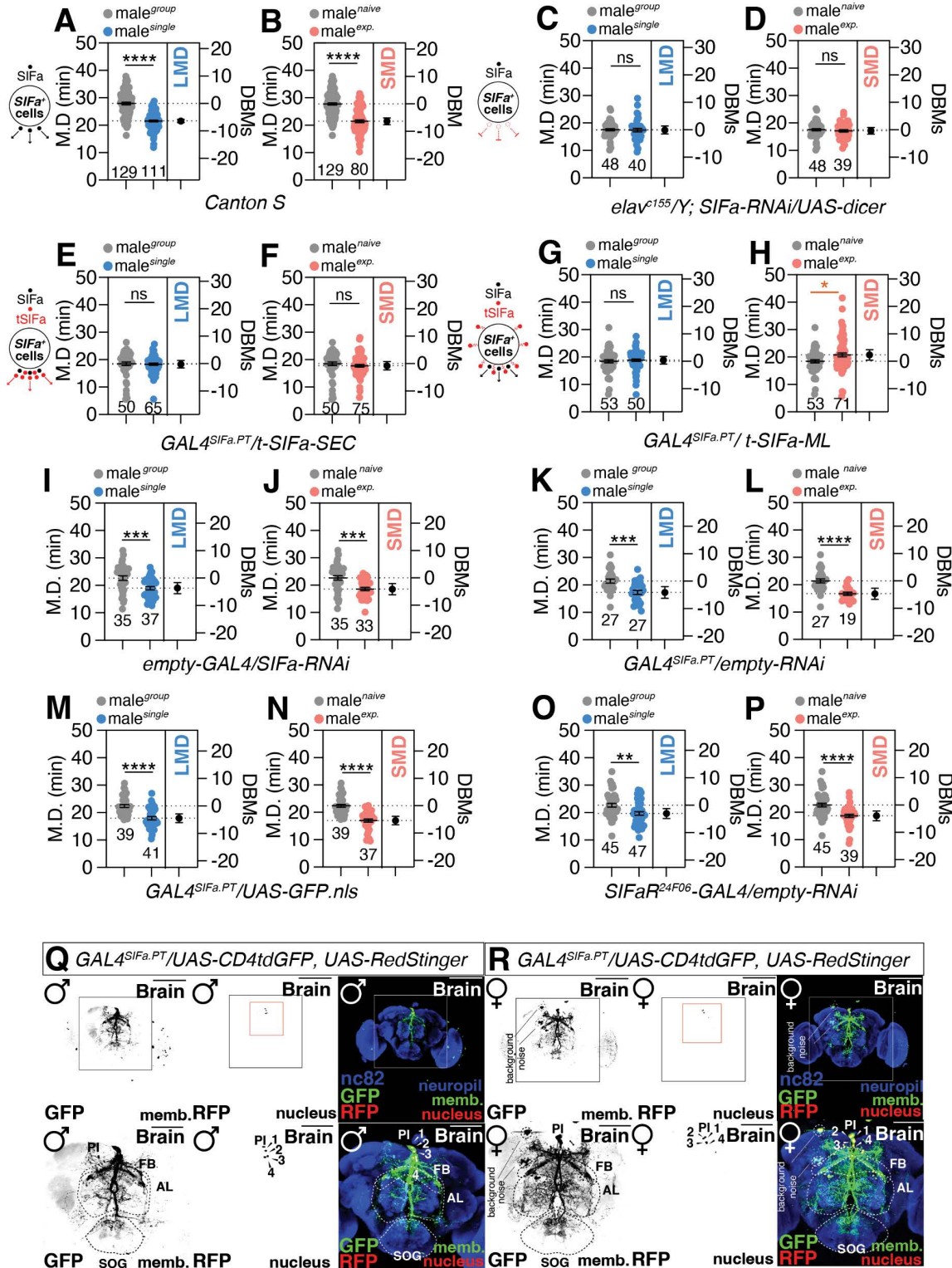

**Fig 1. Neuronal expression of *SIFa* controls interval timing behaviors. (A)** LMD assays of CS males. In the mating duration (MD) assays, light gray data points denote males that were group-reared (or sexually naïve), whereas blue (or pink) data points signify males that were singly reared (or sexually experienced). The dot plots represent the MD of each male fly. The mean value and standard error are labeled within the dot plot (black lines).

Asterisks represent significant differences, as revealed by the unpaired Student $t$ test, and ns represents non-significant differences M.D represent mating duration. DBMs represent the 'difference between means' for the evaluation of estimation statistics (See Materials and methods). Asterisks represent significant differences, as revealed by the Student $t$ test (*$p < 0.05$, **$p < 0.01$, ***$p < 0.001$). Consequently, data points on graphs marked with asterisks indicate that LMD or SMD behaviors remain unaltered or within normal parameters, whereas those labeled with 'ns' signify that LMD or SMD behaviors have been perturbed due to mutations or genetic alterations in the respective strains. For detailed methods, see the Materials and methods for a detailed description of the MD assays used in this study. In the framework of our investigation, the routine application of internal controls is employed for the vast majority of experimental procedures, as delineated in the "**Mating Duration Assay**" and "**Statistical Tests**" subsections of the Materials and methods section. The same notations for statistical significance are used in other figures. **(B)** SMD assays of CS males. Light gray dots represent naïve males and pink dots represent experienced ones (two-tailed unpaired $t$ test). In all plots and statistical tests. Data are presented as mean ± s.e.m. ns = not significant ($p > 0.05$), *$p < 0.05$, **$p < 0.01$, ***$p < 0.001$, ****$p < 0.0001$. Sample sizes ($n$) are indicated in the figure panels. **(C and D)** MD assays for $GAL4$ mediated knockdown of SIFa via $SIFa\text{-}RNAi$ using the $elav^{c155}$ driver (two-tailed unpaired $t$ test). In all plots and statistical tests. Data are presented as mean ± s.e.m. ns = not significant ($p > 0.05$), *$p < 0.05$, **$p < 0.01$, ***$p < 0.001$, ****$p < 0.0001$. Sample sizes ($n$) are indicated in the figure panels. **(E and H)** MD assays for GAL4-driven over expression of $SIFa$ via $UAS\text{-}t\text{-}SIFa\text{-}SEC$(E-F) and $UAS\text{-}t\text{-}SIFa\text{-}ML$(G-H) (two-tailed unpaired $t$ test). In all plots and statistical tests. Data are presented as mean ± s.e.m. ns = not significant ($p > 0.05$), *$p < 0.05$, **$p < 0.01$, ***$p < 0.001$, ****$p < 0.0001$. Sample sizes ($n$) are indicated in the figure panels. **(I and J)** MD assays for $empty\text{-}GAL4$ mediated knockdown of $SIFa$ via $SIFa\text{-}RNAi$ (two-tailed unpaired $t$ test). In all plots and statistical tests. Data are presented as mean ± s.e.m. ns = not significant ($p > 0.05$), *$p < 0.05$, **$p < 0.01$, ***$p < 0.001$, ****$p < 0.0001$. Sample sizes ($n$) are indicated in the figure panels. **(K and L)** MD assays for $GAL4^{SIFa\text{-}PT}$ mediated knockdown via $empty\text{-}RNAi$ (two-tailed unpaired $t$ test). In all plots and statistical tests. Data are presented as mean ± s.e.m. ns = not significant ($p > 0.05$), *$p < 0.05$, **$p < 0.01$, ***$p < 0.001$, ****$p < 0.0001$. Sample sizes ($n$) are indicated in the figure panels. **(M and N)** MD assays for $GAL4^{SIFa\text{-}PT}$ expressed with $UAS\text{-}GFP.nls$ (two-tailed unpaired $t$ test). In all plots and statistical tests. Data are presented as mean ± s.e.m. ns = not significant ($p > 0.05$), *$p < 0.05$, **$p < 0.01$, ***$p < 0.001$, ****$p < 0.0001$. Sample sizes ($n$) are indicated in the figure panels. **(O and P)** MD assays for $SIFaR^{24F06}\text{-}GAL4$ mediated knockdown via $empty\text{-}RNAi$ (two-tailed unpaired $t$ test). In all plots and statistical tests. Data are presented as mean ± s.e.m. ns = not significant ($p > 0.05$), *$p < 0.05$, **$p < 0.01$, ***$p < 0.001$, ****$p < 0.0001$. Sample sizes ($n$) are indicated in the figure panels. **(Q and R)** Male (left) and female (right) flies expressing $GAL4^{SIFa\text{-}PT}$ together with $UAS\text{-}CD4tdGFP$ and $UAS\text{-}RedStinger$ were immunostained with anti-GFP (green), anti-RFP (red), nc82 (blue) antibodies. areas outlined by white boxes are enlarged in the bottom panel. Scale bars represent 100 µm. Underlying data for all graphs can be found in file S1 Data.

## Architecture of SIFa-positive neurons in the central nervous system

Four SIFa neurons located in the PI are likely the most extensively branching peptidergic neurons in *Drosophila* [49]. We used an interactive tool for neuron connectivity (Virtual Fly Brain) [50] to precisely determine the location of SIFa-positive neurons. We discovered that the cell bodies of SIFa neurons arborize from the posterior to the anterior region (S1K Fig). It was verified through 3D reconstruction that two dorsal cell bodies are situated further anteriorly than two ventral cell bodies (S1 Movie). Anterior-dorsal SIFa neurons (SIFa$^{DA}$) and posterior-ventral SIFa neurons (SIFa$^{VP}$) could potentially have distinct functions (S1E Fig).

The established connections and architecture of SIFa neurons have been described by Martelli and colleagues, which enhances our understanding of their functional roles within the neuronal circuitry [51]. To identify the dendritic and axonal components of SIFa-neuronal processes, we employed a similar approach to that reported by Martelli [51]. We co-expressed *UAS-Denmark* (a dendritic marker) with *UAS-syt.eGFP* (a presynaptic marker) to label postsynapses and presynapses of neurons labeled by *GAL4* [52]. We discovered that dendrites of SIFa neurons span just the CB area, including the PI, AL, and posterior-ventral region of the protocerebrum (PRW), whereas presynaptic terminals cover nearly the whole brain region (magnified regions in S3A Fig). Colocalization analysis in the total brain (S3B–S3D Fig), the CB (S3E–S3G Fig), and the OL (S3H–S3J Fig) confirms the broader arborization of presynaptic terminals than dendrites.

Within the VNC, the majority of branching structures are found in the abdominal ganglion (AG) (S3K Fig). Additionally, most of these branching structures are presynaptic terminals, which are different from those found in the brain (S3L–S3N Fig). This suggests that the majority of SIFa$^{+}$ neuronal structures in VNC are axons rather than dendrites. Compared to brain SIFa$^{+}$ dendrites, which overlap presynaptic terminals by almost 80%, VNC SIFa$^{+}$ dendrites only overlap presynaptic terminals by around 2%. (S3M and S3N Fig), suggesting that SIFa$^{+}$ cells form extensive synapses each other within the brain but not in the VNC (S3O Fig). We noticed that the $SIFa^{2A\text{-}GAL4}$ strains, which were created through the knock-in (KI) lines using the site-specific integration system, exhibit robust expression exclusively in the brain region (S3P Fig). This indicates the potential existence of distinct arborization and functional differences between the dorsal and ventral SIFa neurons (S1E Fig), which may play a role in modulating physiology and behaviors.

## Glutamatergic and dopaminergic presynaptic inputs from SIFa-positive neurons influence interval timing behavior in a distinct manner from sleep behavior

It has been proposed that SIFa neurons might be able to utilize glutamate, a neurotransmitter, for controlling sleep [46]; however, no comprehensive analysis of neurotransmitter screening has been conducted yet. To determine which neurotransmitter (NT) was responsible for the LMD/SMD, we utilized RNAi-mediated knockdown of key genes which are necessary to produce each NT. We chose *VGlut-RNAi* (*Vesicular glutamate transporter*) to disrupt the glutamatergic (Fig 2A and 2B), *ple-RNAi* (*pale*) for the dopaminergic (DA) (Fig 2C and 2D), *Gad1-RNAi* (*Glutamic acid decarboxylase 1*) for the GABAergic (Fig 2E and 2F), *Trhn-RNAi* (*Tryptophan hydroxylase neuronal*) for the serotonergic (Fig 2G and 2H), *ChAT-RNAi* (*Choline acetyltransferase*) for the cholinergic (Fig 2I and 2J), and *Hdc-RNAi* (*Histidine decarboxylase*) for the histaminergic (Fig 2K and 2L).

By using RNAi-mediated knockdown experiments, we found that glutamatergic and dopaminergic transmission are crucial for generating SMD but not LMD behavior in SIFa-positive neurons (Fig 2A–2L). SIFa also modulates the courtship behavior of male flies [42]. Studies have shown that social isolation can reduce male courtship behavior in *Drosophila melanogaster* [53–55], however, courtship behavior does not change in response to sexual experience [34]. The knockdown of *VGlut* augmented the courtship activity of males reared in isolation, while diminishing it in males with prior sexual experience (S4A–S4D Fig). The removal of *ple*, on the other hand, enhances courtship behavior in both socially isolated and sexually experienced males (S4E and S4F Fig).

Notably, the sleep patterns and overall sleep duration of male flies were not influenced by the interruption of glutamatergic synaptic transmission from SIFa neurons (Fig 2M). The findings indicate that glutamatergic inputs specifically influence SMD behavior, but do not have an effect on sleep. On the other hand, dopaminergic inputs from SIFa neurons have an impact on both sleep and SMD behavior which is consistent with previous research (Fig 2M) [56,57]. To further substantiate the role of glutamate in SIFa-mediated behaviors. We targeted knockdown of VGlut receptors in SIFaR-expressing neurons. Strikingly, the knockdown of VGlut receptors in these neurons also disrupted SMD behavior, mirroring the phenotype observed upon direct suppression of glutamatergic signaling in SIFa neurons (S4G–S4L Fig). We also employed a conditional glutamatergic synaptic vesicle marker to confirm the presence of glutamatergic SIFa neurons (Figs 2P and S4M) [58]. This suggests that glutamate is an essential neurotransmitter for modulating interval timing in SIFa neurons.

Fly Scope single-cell RNA sequencing data suggest potential high-level co-expression of *VGlut* and *ple* in SIFa neurons (Fig 2N and 2O). To verify the presence of dopaminergic (DA) neurons within the SIFa neuron population, we utilized the Dop1R1-Tango system, which produces a GFP signal upon dopamine binding [59]. Our results confirm the existence of active dopamine receptors in SIFaR neurons and demonstrate that these neurons are responsive to dopamine (Fig 3A; genetic control shown in S5I Fig). Given that *SIFaR*$^{24F06}$ neurons have been identified as forming direct synaptic connections with SIFa neurons (Fig 3A), these findings substantiate that SIFa neurons employ DA signaling to modulate SIFaR-expressing neural circuits [60]. Additionally, we conducted RNAi experiments targeting Dop1R1, Dop1R2, DopEcR, and Dop2R receptors in SIFaR-expressing cells to elucidate the role of DA signaling in modulating interval timing mediated by SIFa neurons (S5A–S5H Fig). Our findings indicated that *Dop1R1-RNAi* selectively influenced SMD behavior. Conversely, the other dopamine receptors, when diminished, affected both SMD and LMD behaviors. These results indicate that diverse dopamine receptors have specific functions in modulating interval timing behaviors. We also employed a well-established genetic toolkit to analyze DA circuits and confirmed that a subset of SIFa neurons is dopaminergic (S5J and S5K Fig) [61]. This finding aligns with predictions from Flywire data, which indicate that SIFa neurons include dopaminergic neurons [62,63].

To further validate our hypothesis that SIFa neurons utilize the DA system for neurotransmission, we employed next-generation GRAB (GPCR-activation-based) sensors specifically developed for monitoring dopaminergic activity in vivo [64,65]. These sensors represent a significant advancement in neurotransmitter detection, offering high sensitivity, selectivity, and signal-to-noise properties with subsecond response kinetics and the ability to detect a wide range of dopamine

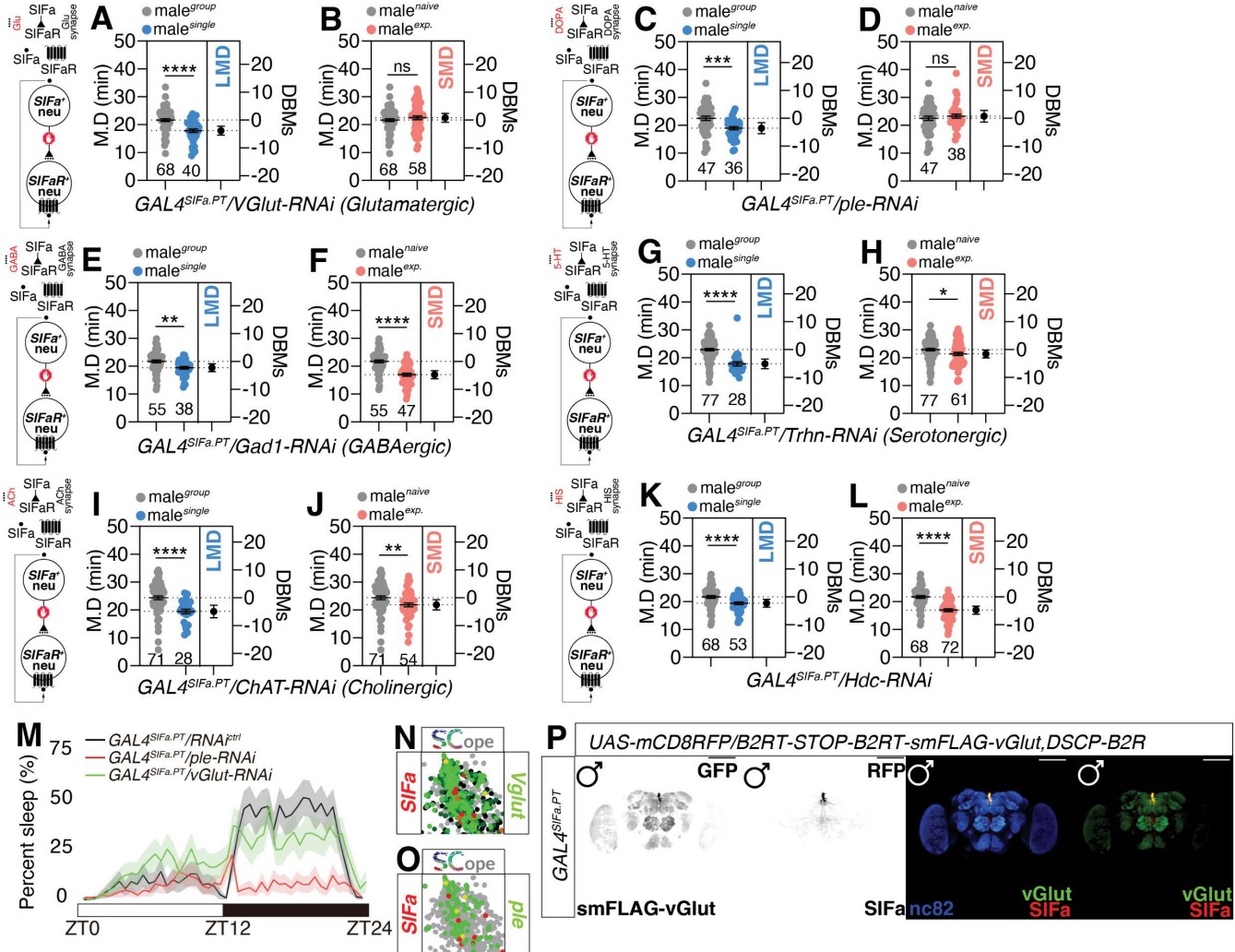

**Fig 2. Interval time is affected by glutaminergic and dopaminergic presynaptic inputs from SIFa-positive neurons. (A–L)** MD assays of flies expressing the *GAL4^SIFa.PT^* driver together with **(A, B)** *VGlut-RNAi* (C, D) *ple-RNAi* **(E, F)** *Gad1-RNAi* **(G, H)** *Trhn-RNAi* **(I, J)** *ChAT-RNAi* **(K, L)** *Hdc-RNAi* (two-tailed unpaired *t* test). In all plots and statistical tests. Data are presented as mean ± s.e.m. ns = not significant ($p > 0.05$), *$p < 0.05$, **$p < 0.01$, ***$p < 0.001$, ****$p < 0.0001$. Sample sizes (*n*) are indicated in the figure panels. The diagram illustrating the correlation between SIFa neurons, SIFaR, and the neurotransmitter is located on the left side. **(M)** Group sleep of flies expressing *GAL4^SIFa.PT^* driver together with *ple-RNAi* (red) less than heterozygous controls (gray) during the course of a 24-h day, but flies together with *VGlut-RNAi* (green) are not. (*n* = 100 flies per group). White and black bars denote periods of light and darkness. **(N, O)** Fly SCope single-cell RNA sequencing data of cells co-expressing *SIFa* together with *VGlut* and *ple*. Each tSNE visualization depicts the coexpression patterns of genes, with each color corresponding to the genes listed on the left, right, and/or bottom of the plot. The tissue name, as referenced on the Fly SCope website is indicated in the upper left corner of the tSNE plot. Consistency in the tSNE plot visualization is preserved across all figures. See the Materials and methods for a detailed description of the single-nucleus RNA-sequencing analyses and code availability used in this study. **(P)** Fly's brain After expression of the B2 recombinase in neurons of SIFa, with a binary transcription system driver and a compatible B2 recombinase responder transgene, the STOP cassette is excised and *smFLAG-vGlut* is expressed in SIFa neurons, above flies with *UAS-mCD8RFP* were immunostained with anti-DsRed (red), anti-FLAG (green) and nc82 (blue) antibodies. Scale bars represent 100 mm. Underlying data for all graphs can be found in file S1 Data.

concentrations [64,65]. We implemented a targeted experimental approach using the *UAS-P2X2* system to achieve precise activation of SIFa neurons through ATP application [66,67]. This system enables specific neuronal activation while maintaining temporal precision, as P2X2 receptors are ATP-gated ion channels that generate rapid, nonspecific cation conductance

upon ATP binding. Our experimental design involved expressing GRAB sensors in SIFaR neurons while simultaneously expressing P2X2 receptors in SIFa neurons. Upon ATP application to activate the P2X2-expressing SIFa neurons, we monitored the resulting DA signals through the GRAB sensor responses in downstream SIFaR neurons. As predicted by our previous experimental evidence supporting the presence of dopaminergic components within SIFa neurons, the GRAB sensors exhibited activation in SIFaR neurons following ATP-induced stimulation of SIFa neurons (Fig 3B–3D).

This response pattern provides compelling evidence for several key aspects of SIFa neurotransmission: First, it confirms functional dopamine release from SIFa neurons upon activation. Second, it demonstrates the existence of functional dopaminergic signaling pathways from SIFa to SIFaR neurons. Third, it validates the physiological relevance of dopaminergic co-transmission in SIFa neurons, complementing their known neuropeptidergic signaling. Our demonstration that SIFa neurons can release dopamine positions them as critical modulators in timing circuits, capable of transmitting internal state information that influences context-dependent interval timing behaviors. All these data indicate that the modulation of diverse behavioral repertoires can be achieved by the use of various combinations of neurotransmitter co-transmission in SIFa neurons.

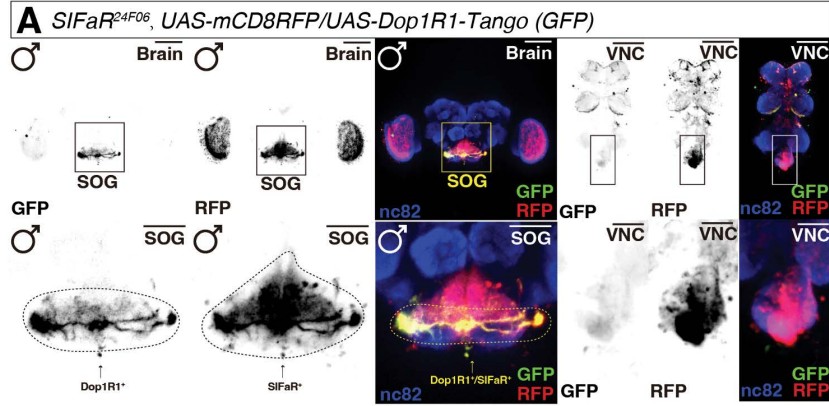

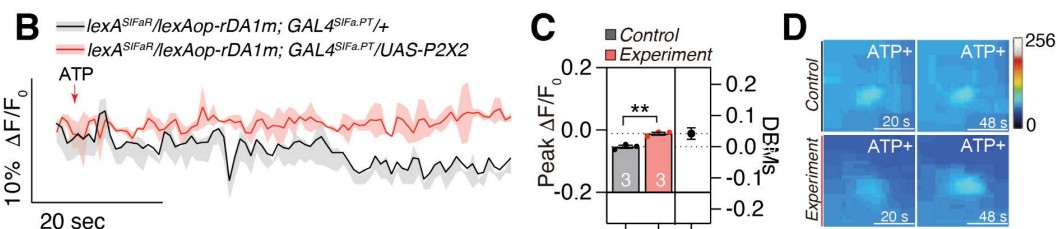

**Fig 3. Dopamine release from SIFa neurons detected by dopamine sensor. (A)** male flies expressing *UAS-Dop1R1-Tango* and *SIFaR^24F06* drivers together with *UAS-mCD8RFP* were imaged under a fluorescent microscope with anti-GFP (green), anti-DsRed (red), and nc82 (blue) antibodies. yellow parts showed that *SIFaR^24F06* neurons were activated by dopamine, with the co-localization of *Dop1R1-Tango*-GFP and *SIFaR^24F06*-mCD8RFP in the SOG region demonstrating the presence of dopaminergic input to SIFaR neurons. Scale bars represent 100 μm. Areas outlined by boxes are enlarged in the bottom panel. Scale bars represent 50 μm. **(B)** Representative fluorescence images of rDA1m expressed in SIFaR cells. Gray lines as genetic control for red. **(C)** Peak fluorescence changes ($\Delta F/F$) in SIFaR neurons of *LexAop-rDA1m, UAS-P2X2* with *LexA^SIFaR*, *GAL4^SIFa.PT* driver upon ATP application (red, $n = 3$) compared with flies expressing *LexAop-RDA1m* with *LexA^SIFaR*, *GAL4^SIFa.PT* driver (gray as control, $n = 3$). DBMs represent the 'difference between means' for the evaluation of estimation statistics (See Materials and methods) (two-tailed unpaired $t$ test). In all plots and statistical tests. Data are presented as mean ± s.e.m. ns = not significant ($p > 0.05$), *$p < 0.05$, **$p < 0.01$, ***$p < 0.001$, ****$p < 0.0001$. Sample sizes ($n$) are indicated in the figure panels. **(D)** signal response of SIFaR cell bodies after application of ATP. Scale bar represented 10 μm. Underlying data for all graphs can be found in file S1 Data.

## sNPF inputs are critical to generate SIFa-mediated internal states for interval timing behavior

The short neuropeptide F (sNPF) is involved in executive signaling inside local neuroendocrine cells. It functions as a co-transmitter through its receptor, sNPF-R, in specific circuits. The sNPF governs the regulation of food intake and body size [68,69], is sleep-promoting inhibitory modulator [70–72], affects olfactory memory formation [73], and control interval timing behavior [33]. sNPF signaling plays a crucial role in regulating feeding behavior in *Drosophila melanogaster*, influencing food intake and body size [44,68,74,75]. However, there is currently no direct evidence reported linking sNPF signaling to SIFa neurons.

Knockdown of sNPF in SIFa-expressing neurons (Fig 4A and 4B) or SIFa in sNPF-expressing cells (Fig 4C and 4D) has no impact on both LMD and SMD behavior, implying that both neuropeptide-expressing cells are mutually exclusive. Knockdown of sNPF-R in SIFa-expressing cells (Fig 4E and 4F) or SIFa in sNPF-R-expressing cells (Fig 4G and 4H) disrupts only SMD behavior, implying that sNPF-R function in SIFa-positive cells and SIFa function in sNPF-R-positive cells are required to generate SMD. SIFa transmits information to adjacent cells through the SIFaR. Inhibition of SIFaR in cells expressing sNPF leads to the disruption of both LMD and SMD behaviors (Fig 4I and 4J). Similarly, the knockdown of sNPF in cells expressing SIFaR also disrupts both behaviors (Fig 4K and 4L). When the expression of SIFaR was suppressed in cells expressing sNPF-R using *sNPF-R^64H09^-GAL4* drivers, only the SMD behavior was affected (S6A and S6B Fig). However, when the expression of sNPF-R was suppressed in SIFaR-expressing cells, both LMD and SMD behaviors were disrupted (S6C and S6D Fig). These data with genetic control experiments (Fig 4M–4R) indicate that the signaling of SIFa-SIFaR is regulated by sNPF-sNPF-R signaling, and that SIFa and sNPF collaborate to regulate interval timing behaviors in a specific manner.

The coexpression of SIFa with sNPF-R was verified by genetic intersection approaches (Figs 4T, S6E–S6G) and fly SCope data (S6H Fig). The extensive coexpression patterns of sNPF with SIFaR were also validated using the same methodology (S6F and S6I Fig). Upon performing GRASP using *SIFa^2A-lexA^* in conjunction with *sNPF-R^64B11^-GAL4*, we observed the formation of robust synapses in the PI tract and the lateral accessory lobe (LAL) area connecting them. Interestingly, when males are exposed to social isolation or sexual experience, there is an increase in synaptic signals around the PI region where SIFa cells are located (S6J Fig). This indicates that the synapses between SIFa cells expressing sNPF-R become stronger (S6K–S6M Fig) since we have found that SIFa cells express sNPF-R (Figs 4T, S6E and S6G). In contrast, we found no synapses between *SIFa^2A^* and *sNPF-R^64H09^* in VNC (S6J Fig). These findings demonstrate that the synaptic plasticity among SIFa cells in the brain plays a crucial role in regulating interval timing behaviors.

To investigate the response of SIFa neurons following the activation of sNPF neurons in the brain, we used the ATP/P2X2 system [76] in an ex vivo preparation to stimuli sNPF cells and simultaneously recorded $Ca^{2+}$ influx in SIFa using GCaMP6 [77] (Fig 4U–4W). To selectively stimulate SIFa neurons, the adenosine triphosphate (ATP)-gated cation channel P2X2 was expressed in sNPF neurons to allow specific depolarization by ATP application, expressed using the *sNPF^2A^-GAL4* driver. The $Ca^{2+}$ fluctuations were tracked in these neurons using fluorescent microscopy in conjunction with a genetically encoded calcium indicator, GCaMP6, which was expressed under the control of the *LexA^SIFa.PT^* driver. Upon activation of sNPF cells via 1mM ATP perfusion, strong $Ca^{2+}$ responses were elicited in SIFa cells (Fig 4U–4W). A similar timescale of $Ca^{2+}$ responses has been observed in the temperature coding of central circadian circuits [78] but is different from the time scale of $Ca^{2+}$ responses of glucose-sensing neurons in the brain [79]. Therefore, our GCaMP responses align with the neuronal signaling cascades occurring within brain interneuron circuits. In contrast, no change in GCaMP fluorescence was detected in the control group treated with artificial hemolymph (AHL) buffer. These findings demonstrate that sNPF neuronal activation strongly activates SIFa neurons.

The cells that express both sNPF and SIFaR are predominantly found in the AG (shown by yellow dotted lines labeled as "cell" in S6F Fig). This indicates that the sNPF-positive cells responsible for SIFa-SIFaR signaling to induce SMD are located in the VNC rather than the brain. The results suggest that SIFa signals originating from the brain and sNPF signals originating from the VNC create a feedback or feedforward loop to modulate the internal states of the CNS in male

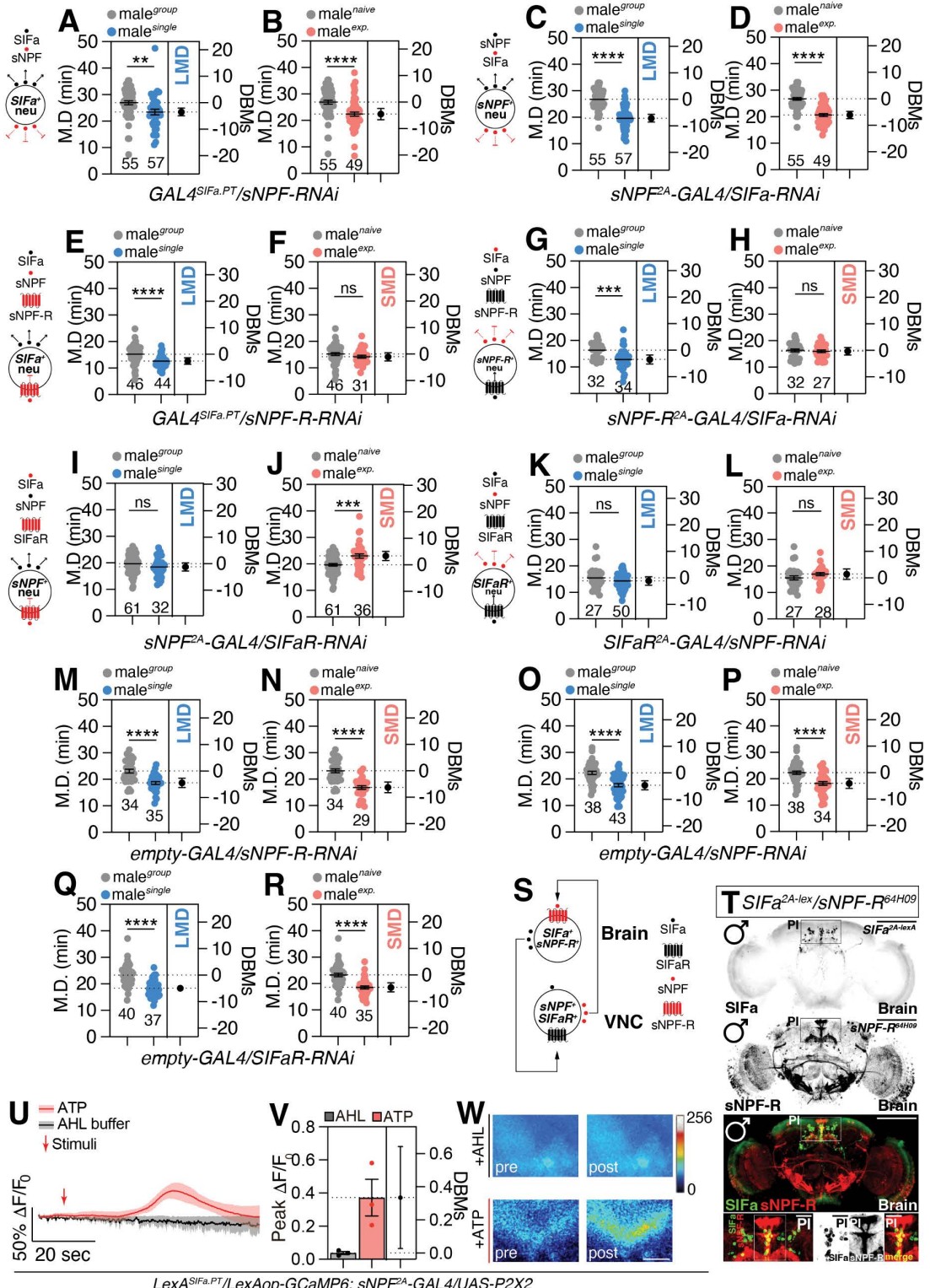

**Fig 4. sNPF signals are necessary to create internal states mediated by SIFa. (A, B)** MD assays of flies expressing the *GAL4^SIFa.PT^* driver together with *sNPF-RNAi* (two-tailed unpaired *t* test). In all plots and statistical tests. Data are presented as mean ± s.e.m. ns = not significant (*p* > 0.05), *p < 0.05, **p < 0.01, ***p < 0.001, ****p < 0.0001. Sample sizes (*n*) are indicated in the figure panels. **(C, D)** MD assays of flies expressing the *sNPF^2A^-GAL4* driver

together with *SIFa-RNAi* (two-tailed unpaired *t* test). In all plots and statistical tests. Data are presented as mean ± s.e.m. ns = not significant ($p > 0.05$), *$p < 0.05$, **$p < 0.01$, ***$p < 0.001$, ****$p < 0.0001$. Sample sizes (*n*) are indicated in the figure panels. **(E, F)** MD assays of flies expressing the *GAL4^SIFa.PT^* driver together with *sNPF-R-RNAi* (two-tailed unpaired *t* test). In all plots and statistical tests. Data are presented as mean ± s.e.m. ns = not significant ($p > 0.05$), *$p < 0.05$, **$p < 0.01$, ***$p < 0.001$, ****$p < 0.0001$. Sample sizes (*n*) are indicated in the figure panels. **(G, H)** MD assays of flies expressing the *sNPF-R^2A^-GAL4* driver together with *SIFa-RNAi* (two-tailed unpaired *t* test). In all plots and statistical tests. Data are presented as mean ± s.e.m. ns = not significant ($p > 0.05$), *$p < 0.05$, **$p < 0.01$, ***$p < 0.001$, ****$p < 0.0001$. Sample sizes (*n*) are indicated in the figure panels. **(I, J)** MD assays of flies expressing the *sNPF^2A^-GAL4* driver together with *SIFaR-RNAi* (two-tailed unpaired *t* test). In all plots and statistical tests. Data are presented as mean ± s.e.m. ns = not significant ($p > 0.05$), *$p < 0.05$, **$p < 0.01$, ***$p < 0.001$, ****$p < 0.0001$. Sample sizes (*n*) are indicated in the figure panels. **(K, L)** MD assays of flies expressing the *SIFaR^2A^-GAL4* driver together with *sNPF-RNAi* (two-tailed unpaired *t* test). In all plots and statistical tests. Data are presented as mean ± s.e.m. ns = not significant ($p > 0.05$), *$p < 0.05$, **$p < 0.01$, ***$p < 0.001$, ****$p < 0.0001$. Sample sizes (*n*) are indicated in the figure panels. **(M–R)** MD assays for *GAL4* mediated knockdown of *sNPF-R* (M, N), *sNPF* (O, P), *SIFaR* (Q, R) using *empty-GAL4* driver (two-tailed unpaired *t* test). In all plots and statistical tests. Data are presented as mean ± s.e.m. ns = not significant ($p > 0.05$), *$p < 0.05$, **$p < 0.01$, ***$p < 0.001$, ****$p < 0.0001$. Sample sizes (*n*) are indicated in the figure panels. **(S)** Schematic depicting the interaction between SIFa and sNPF neurons. **(T)** male flies expressing *SIFa^2A-lexA^* and *sNPF-R^64H09^-GAL4* drivers together with *lexAop-mCD8RFP* and *UAS-mCD8RFP* were imaged live under a fluorescent microscope with anti-GFP (green), anti-DsRed (red), and nc82 (blue) antibodies. yellow arrows indicate *SIFa*-positive neurons and *sNPF-R*-positive neurons. Scale bars represent 100 μm. The following panel is the enlarged version of the box on PI. **(U)** Activation of sNPF neurons increases SIFa neuronal activity. Representative images of GCaMP fluorescence in SIFa neurons located in the fan-shaped body before and after ATP application (experimental group) or AHL buffer application (control group). **(V)** Peak fluorescence changes (ΔF/F) in SIFa neurons upon ATP application (red, $n = 3$) or AHL buffer application of flies expressing *LexAop-GCaMP6*, *UAS-P2X2* with *LexA^SIFa.PT^*, *sNPF^2A^-GAL4* driver (gray, $n = 3$). DBMs represent the 'difference between means' for the evaluation of estimation statistics (See Materials and methods) (two-tailed unpaired *t* test). In all plots and statistical tests. Data are presented as mean ± s.e.m. ns = not significant ($p > 0.05$), *$p < 0.05$, **$p < 0.01$, ***$p < 0.001$, ****$p < 0.0001$. **(W)** Calcium response of SIFa cell bodies in the fan-shaped body before (left) and after (right) application of AHL (control) or ATP. Scale bar represented 40 μm. Underlying data for all graphs can be found in file S1 Data.

flies when they are exposed to varying social environments. The internal states created by the interaction of these two neuropeptides and their receptors, which signal through the brain to VNC, are crucial components of the circuit responsible for interval timing behaviors (Fig 4S).

## The neuronal activity and synaptic plasticity of SIFa neurons determine internal states for interval timing behavior

The activity of SIFa neurons can be changed by a number of different stimuli [44,49]. However, the precise mechanisms by which they regulate SIFa neuronal activity remain unknown. To determine whether neuronal activities undergo alterations in SIFa neurons associated with interval timing, we utilized the TRIC (transcriptional reporters for sensing $Ca^{2+}$) system [80]. TRIC serves as a valuable complement to functional $Ca^{2+}$ imaging by integrating long-term changes in neuronal activity and providing genetic access to neurons based on their activity. Given that both LMD and SMD require at least 6–12 h of social isolation or sexual interaction [29,81], repeated sensory inputs could potentially lead to the accumulation of the modified transcription factor within the nucleus of activated neurons in vivo. Indeed, social isolation or sexual experiences affected the neural activity of *GAL4^SIFa.PT^*-labeled neurons (Fig 5A). Male flies with social isolation or sexual experience exhibited robust TRIC fluorescence in the PI region (Fig 5B and 5C). In contrast, the TRIC fluorescence in the prow (PRW) region becomes weakened in males raised in isolation and stronger in males with sexual experience (Fig 5D and 5E). Considering the similarity of the control RFP signals across conditions, we conclude that social isolation and sexual experience alter $Ca^{2+}$ levels in SIFa neurons in a comparable yet distinct manner (Fig 5J).

Next, to determine if the temporal activation of SIFa neurons may generate distinct internal states for MD in the absence of social isolation of sexual experience, we expressed the heat-sensitive *Drosophila* cation channel TrpA1 in SIFa neurons and then transferred the experimental group to the activation temperature (29 °C). Surprisingly, flies expressing TrpA1 in SIFa neurons at the activation temperature exhibited a SMD than flies maintained at 22 °C (Fig 5F) compared to genetic control (Fig 5H and 5I). However, the temporal activation of SIFa neurons did not have an impact on courtship behavior (Fig 5G). This indicates that the MD and the level of courtship activity are controlled separately by the activity of SIFa neurons. Stimulation of SIFa neurons resulted in an elevation in food consumption, as assessed using the EX-Q (excreta quantification)

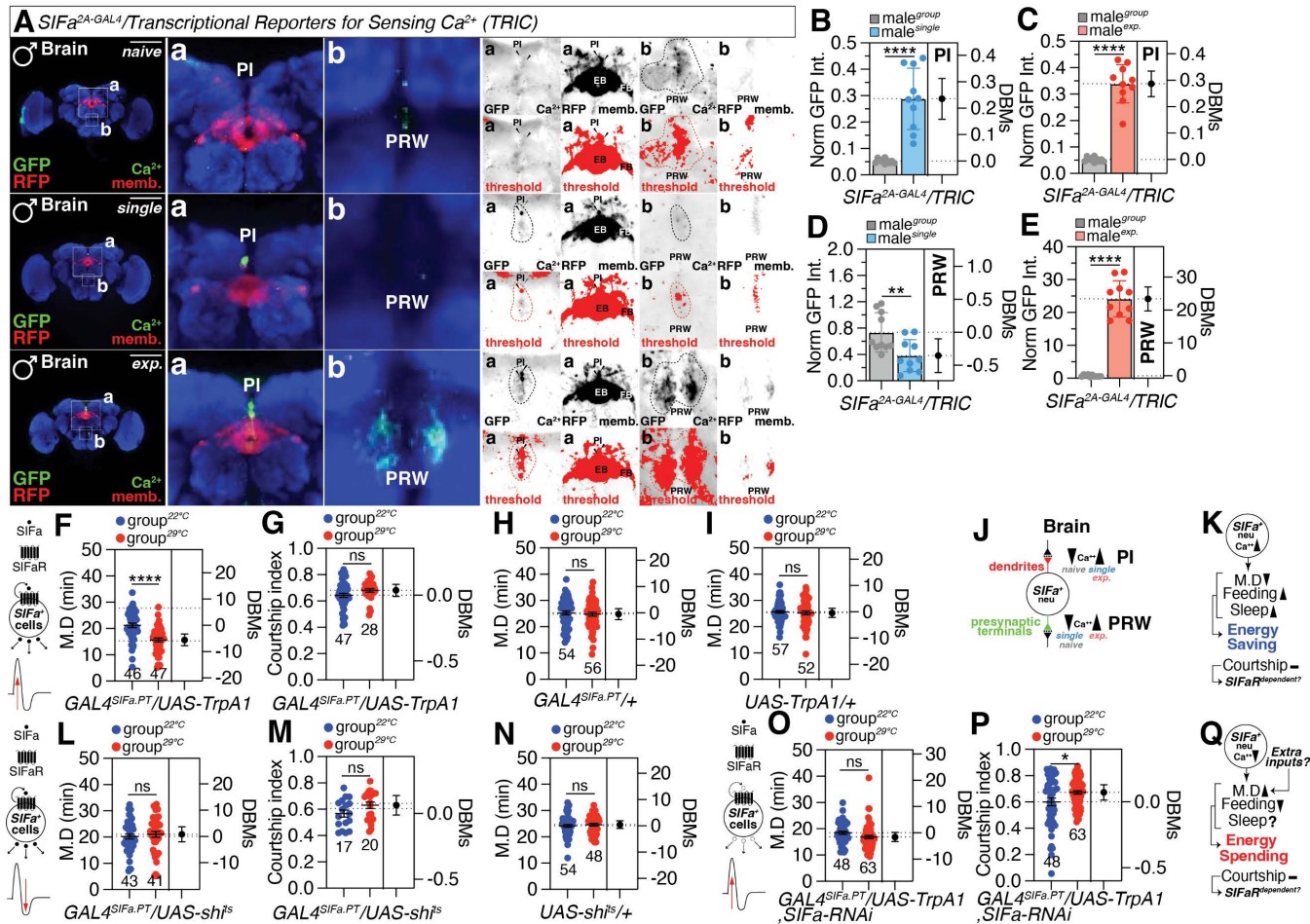

**Fig 5. Internal interval timing behavior is determined by *SIFa* neuronal activity and synaptic plasticity. (A)** CaM/MKII-mediated TRIC signal in the brain of transgenic flies (*nsyb-MKII::GAL4DBDo, UAS-p65AD::CaM, 10XUAS-IVS-mCD8::RFP, 13XLexAop2-mCD8::GFP*). The right four panels are presented as a gray scale to clearly show the TRIC signal of *SIFa* neurons in PI and PRW labeled by *SIFa²ᴬ⁻ᴳᴬᴸ⁴* driver and TRIC. Scale bars represent 100 μm in left panels and 10 μm in a and b. **(B–E)** Quantification of GFP signal of male fly expressing *SIFa²ᴬ⁻ᴳᴬᴸ⁴* drivers together with Transcriptional Reporters for Sensing Ga²⁺ of PI and PRW region. 'Norm. GFP Int.' refers to the normalized GFP intensity relative to the RFP signal (two-tailed unpaired *t* test). In all plots and statistical tests. Data are presented as mean±s.e.m. ns=not significant (*p*>0.05), *p<0.05, **p<0.01, ***p<0.001, ****p<0.0001. **(F-G, L-M)** Temperature-shift mating duration and courtship index of male fly expressing *GAL4ˢᴵᶠᵃ.ᴾᵀ* drivers together with heat-sensitive *Drosophila* cation channel *TrpA1* and *shiᵗˢ* (two-tailed unpaired *t* test). In all plots and statistical tests. Data are presented as mean±s.e.m. ns=not significant (*p*>0.05), *p<0.05, **p<0.01, ***p<0.001, ****p<0.0001. Sample sizes (*n*) are indicated in the figure panels. **(H, I and N)** Genetic control of MD assay of flies expressing *TrpA1*(I) and *shiᵗˢ* (N) in *SIFa* neurons at the activation and deactivation temperature (two-tailed unpaired *t* test). In all plots and statistical tests. Data are presented as mean±s.e.m. ns=not significant (*p*>0.05), *p<0.05, **p<0.01, ***p<0.001, ****p<0.0001. Sample sizes (*n*) are indicated in the figure panels. **(J)** Alterations in Ca²⁺ concentrations within SIFa neurons in response to social isolation and sexual experience exhibit a comparable yet distinct profile. **(O, P)** Analysis of mating duration and courtship index in male *Drosophila* expressing *GAL4ˢᴵᶠᵃ.ᴾᵀ* drivers, in conjunction with the heat-sensitive *Drosophila* cation channel *TrpA1* and *SIFa-RNAi*, following temperature shifts (two-tailed unpaired *t* test). In all plots and statistical tests. Data are presented as mean±s.e.m. ns=not significant (*p*>0.05), *p<0.05, **p<0.01, ***p<0.001, ****p<0.0001. Sample sizes (*n*) are indicated in the figure panels. **(K and Q)** SIFa neurons modulate energy expenditure through the fine-tuning of their intracellular environments. Underlying data for all graphs can be found in file S1 Data.

method [82] (S7A and S7B Fig) and the colorimetric food intake assay [83] (S7C and S7D Fig). Notably, the temporary deactivation of SIFa neurons through the expression of shiᵗˢ did not have any impact on the MD or the level of courtship activity (Fig 5L and 5N) within the temperature range of 22–29 °C. Nevertheless, the temporary deactivation of SIFa neurons leads

to a decrease in food consumption in male flies (S7E–S7H Fig) as previously described by Martelli and colleagues' report in female flies [44]. These findings indicate that activating SIFa-positive neurons is sufficient to generate specific internal states that lead to reduced MDs and increased food intake. However, it is not sufficient to alter courtship activities (Fig 5K). Moreover, the deactivation of SIFa neurons has the potential of generating internal states that decrease food consumption [51], but it does not have the ability to alter the MD or courtship behavior (Fig 5Q).

We then tried to alter the neuronal activity of SIFa-positive neurons during social isolation and sexual experiences by expressing various modifiers of neuronal activity with *GAL4^{SIFa.PT}*. By expressing TNT (tetanus toxin light chain) to block synaptic transmission in SIFa neurons, we observed that both LMD and SMD were impaired (S7I and S7J Fig). However, food intake was shown to be elevated under these circumstances (S7K–S7L Fig). In contrast to TNT expression, potassium channels KCNJ2/OrkΔC [84] or bacterial sodium channel NachBac expression in SIFa neurons only impaired either LMD or SMD (S7M–S7N; S7U–S7V; S7Q–S7R Fig). Notably, the food intake was partially increased by constitutive modulation of SIFa neuronal activities whether neuronal activity was activated or inhibited (S7O–S7P; S7S–S7T Fig). TNT has been used effectively in several studies to inhibit neurotransmitter and neuropeptide release, KCNJ2/OrkΔC prevents membrane depolarization, and NachBac induces sodium conductance, which causes cell depolarization [85]. Thus, in accordance with genetic control experiments (S8G–S8J Fig), our research suggests that the release of neurotransmitter and neuropeptide vesicles is necessary for the transition of internal states, leading to the production of both LMD and SMD, as well as regulating food intake (S7W Fig).

When TrpA1 artificially stimulated SIFa neurons while simultaneously knocking down SIFa, the MD remained the same (Fig 5O). However, there was a modest increase in courtship activity (Fig 5P). The data support our hypothesis that altering internal states for interval timing activities requires augmentation between SIFa synaptic transmission and SIFa release (S7W Fig).

### Internal states of a socially isolated or sexually experienced male's nervous system represent neuronal plasticity of SIFa neurons

Even though calcium activity successfully represents the internal states that determine the MD of male flies, it is still unknown why similar neuronal alteration affect LMD and SMD in different ways (S7M–S7N; S7Q–S7R; S7U–S7V Fig). Thus, we used *DenMark* and *syt.eGFP* to measure the synaptic plasticity of SIFa neurons in various conditions due to their capacity to detect broad changes in synaptic structure. DenMark signals in the brains of grouped (naïve), singly reared (single), and sexually experienced (exp.) males were comparable; however, syt.eGFP signals in the brains of group reared males were significantly stronger than those in the brains of both singly reared and sexually experienced males (Fig 6A–6E). Similar to the brain, the syt.eGFP signals in the VNC of singly reared and sexually experienced males were significantly weaker than in the VNC of grouped males (Fig 6F–6L). The findings indicate that significant alterations in the postsynaptic region, rather than the presynaptic region, occur in the CNS of males who are in groups, socially isolated, or undergo sexual experience (Fig 6M). These variations in extensive structural synaptic changes may explain why various neuronal activity alterations influence LMD and SMD in different ways.

Finally, we sought to identify circuits downstream of *GAL4^{SIFa.PT}* neurons. At first, we used a recently developed system for unbiased *trans*-synaptic labeling: *trans-Tango*, which induces both myrGFP and mtdTomato to be produced by certain presynaptic neurons and their post-synaptic partners through an engineered signaling pathway [86]. When *trans-Tango* was driven by *GAL4^{SIFa.PT}*, we observed a significant overlap of 40%–45% between SIFa neurons and SIFa *trans*-Tango signals in the PI and AG region (S8A–S8C Fig). These regions are critical for processing sensory inputs and controlling reproductive behaviors, respectively. We hypothesize that the SIFa neurons, through their extensive projections, form a circuitry that translates social experiences into changes in MD. This finding not only suggests that SIFa neurons establish synaptic connections with each other, which supports the results obtained from the DenMark and syt.eGFP experiments (S3O Fig) but also corroborates the synaptic connectivity patterns previously reported [45]

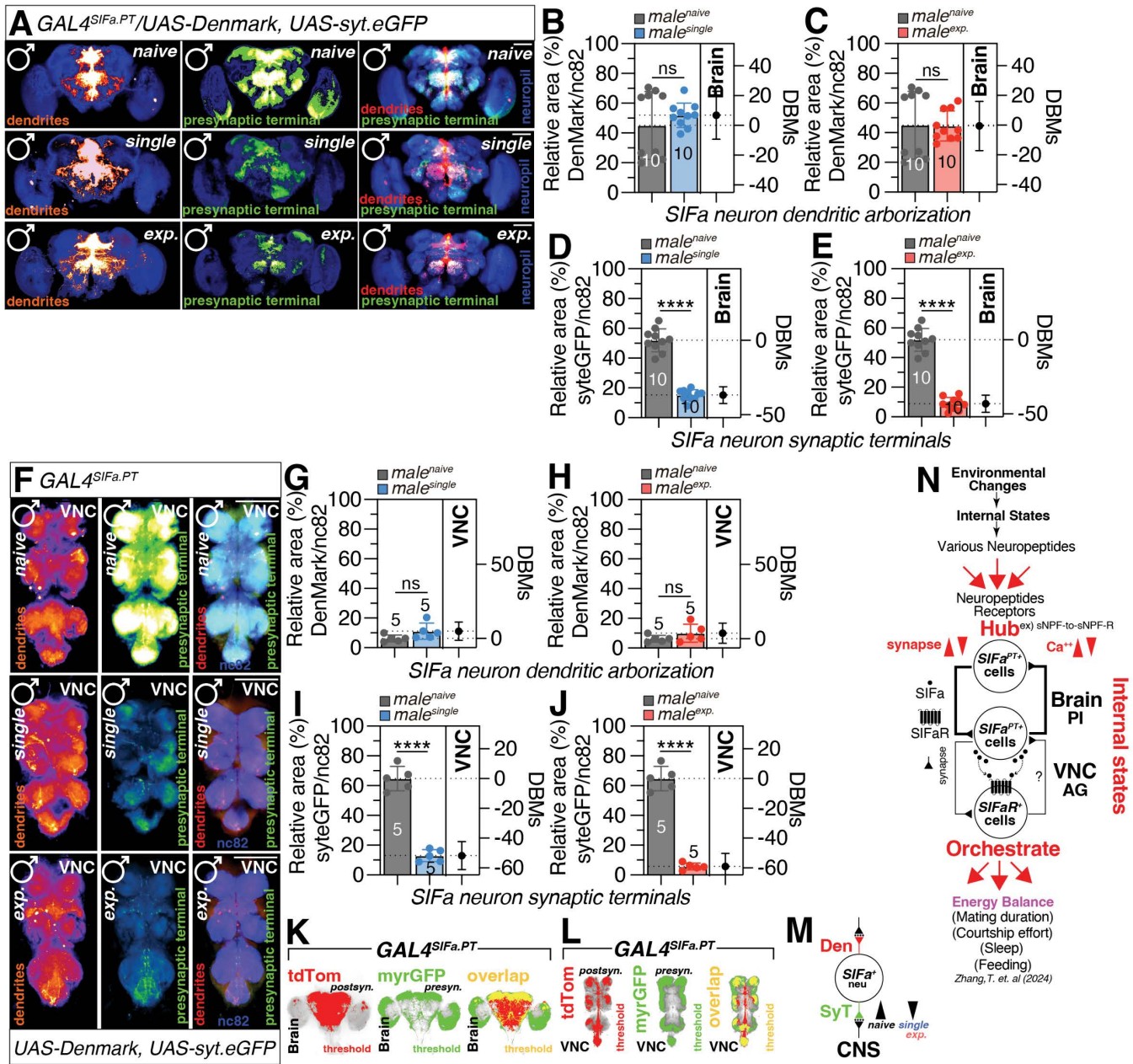

**Fig 6. SIFa neuronal plasticity is seen in socially isolated or sexually experienced males' neural systems. (A)** Image of a brain in which *GAL-4*^SIFa.PT^ was used to simultaneously drive expression of the dendritically-localized Denmark (red) and presynaptic terminal (green) of naïve (top), single (middle) and mated (bottom) male flies. Scale bars represent 100 μm. **(B–E)** Quantification of the relative percent area formed by *GAL4*^SIFa.PT^ with *UAS-DenMark*(red) and *UAS-syt.eGFP*(green) in the brains of male flies was conducted (two-tailed unpaired *t* test). In all plots and statistical tests. Data are presented as mean ± s.e.m. ns = not significant (*p* > 0.05), *p < 0.05, **p < 0.01, ***p < 0.001, ****p < 0.0001. Sample sizes (*n*) are indicated in the figure panels. **(F)** image of a VNC in which *GAL4*^SIFa.PT^ was used to simultaneously drive expression of the dendritically-localized Denmark (red) and presynaptic terminal (green) of naïve (top), single (middle) and mated (bottom) male flies. Scale bars represent 100 μm. **(G–J)** Quantification of dendritic arborization and synaptic terminals of *SIFa* neurons in [Fig 6F](two-tailed unpaired *t* test). In all plots and statistical tests. Data are presented as mean ± s.e.m. ns = not significant (*p* > 0.05), *p < 0.05, **p < 0.01, ***p < 0.001, ****p < 0.0001. Sample sizes (*n*) are indicated in the figure panels. **(K, L)** The threshold of RFP fluorescence (left panel, red), GFP fluorescence (middle panel, green), and overlapping area of GFP and RFP (right panel, yellow) in male fly brain and VNC was marked by threshold function of ImageJ. **(M)** Schematic representation of dendritic arborization and presynaptic terminal dynamics in SIFa neurons subjected to various experimental conditions. **(N)** Schematic depiction of the anatomical arrangement and functional connectivity between SIFa neurons located in the PI of the brain and SIFaR neurons situated in the AG of the VNC, showing how neuropeptide receptor activation regulates energy conservation and spending in *Drosophila* melanogaster. Underlying data for all graphs can be found in file [S1 Data](.)

SIFa neurons project to the protocerebral bridge (PB), which demonstrates several fundamental wiring principles within the central complex (CX), indicating that PB performs crucial computational tasks in the fly brain, such as the transformation of sensory (e.g., visual) input into locomotor commands [87]. SIFa neurons also project to noduli (No) in the CX, which are involved in olfactory processing and memory formation [88,89], and the PRW region, where it is attached to the subesophageal ganglion (SOG), which contains the circuitry underlying feeding behavior and is involved in numerous other aspects of sensory processing and motor control [90]. Additionally, we discovered that SIFa neurons project to a small subset of OL neurons (Bottom panels in S8A Fig). The most prominent projection of SIFa neurons is the AG of the VNC, which regulates abdominal muscles, the gastric system, and reproductive organs [91]. These findings imply that SIFa neurons project to an important brain region for extremely complicated computation and to a crucial VNC region for control of reproductive behavior, all of which modulate interval timing behavior based on the social context (S8F Fig).

To examine the SIFa projections in further detail, we utilized *SIFaR^24F06^-GAL4*, which we recently identified as a key *GAL4* strain, which labels the essential VNC neurons expressing SIFaR for both LMD and SMD in AG [60]. The majority of *SIFaR^24F06^* neurons in the brain project to the PI, SOG, and OL regions (GFP panels of Brain in S8D Fig). In VNC, the majority of *SIFaR^24F06^* neurons project to the AG region RFP signals J (S8D Fig). We discovered that the projection of SIFa neurons in AG parallels the myrGFP signals of *SIFaR^24F06^* neurons, indicating that these AG neurons are the direct synaptic target of SIFa neurons from the brain to the VNC to modulate interval timing. (S8E Fig). Our results suggest that SIFa neurons in the PI of the brain and SIFaR neurons in the AG of the VNC are functionally linked for SIFa-SIFaR signaling (S8F Fig).

## Discusssion

In this study, we conducted a wide search to identify the neuropeptide SIFa that influences both male-specific interval timing behaviors in *D. melanogaster* (S1–S5 Files, S1 Table). We examined the association of SIFa-mediated signaling and its functional relevance to various behaviors including interval timing. The level of SIFa expression in four neurons adjacent to the PI region is essential for the generation of both LMD and SMD behaviors (Figs 1 and S1). SIFa neurons extend their axons, rather than dendrites, throughout the entire brain and VNC, establishing robust synapses exclusively within the brain area (S2 and S3 Fig). SIFa produces several neurotransmitters, including glutamine and dopamine. However, only dopamine is essential for regulating SIFa-induced sleep, while glutamate plays a role in moderating SMD behavior and courtship activity. The sNPF-sNPF-R peptidergic signaling provides input signals to SIFa neurons, which subsequently send back these signals to sNPF neurons through SIFaR. This feedback loop modifies the activity of LMD and SMD in particularly (Figs 4 and S6). The elevated neuronal activity of SIFa neurons is linked to reduced MD as well as increased food intake, and vice versa. This suggests that the modulation of energy homeostasis-related behavioral change is dependent on the activity of SIFa neurons, as well as the co-transmission of octopamine and SIFa following this process (Figs 5 and S7). The elevated neuronal activity of SIFa neurons is closely linked to reduced branching of SIFa axons at the presynaptic level. Furthermore, the postsynaptic neurons of SIFa exhibit a significant degree of overlap in the CB, OL, and AG regions of the nervous system, where SIFaR is expressed (Figs 6 and S8). Therefore, these findings highlight the crucial role of SIFa neuropeptides in regulating various behaviors in *D. melanogaster*, including interval timing, sleep, courtship, and feeding, through complex interactions with other neuropeptidergic systems.

The absence of mating defects following acute synaptic inhibition of SIFa neurons (shi^ts^) contrasted with the impairments observed upon chronic knockdown of *Vglut*, *ple*, or SIFa itself, reveals a fundamental characteristic of SIFa neuron function (Figs 5L–5N versus S7E–S7H). We propose that these neurons employ dual transmission modes: fast synaptic glutamate signaling (mediated by small clear vesicles, SCVs) and activity-independent neuropeptide release (via dense-core vesicles, DCVs). While shibir^ts^ disrupts SCV recycling and acute synaptic transmission [92], it spares DCV-mediated SIFa neuropeptide secretion. Conversely, chronic depletion of glutamate (*Vglut-RNAi*), neuropeptide processing (*ple-RNAi*), or SIFa itself eliminates essential components for sustained neuromodulation. This distinction highlights that MD

regulation depends predominantly on neuropeptide-mediated signaling rather than fast neurotransmission, aligning with established DCV resilience to dynamin-dependent endocytosis blockade [93]. Thus, the phenotype manifests only when the neuropeptide pathway is chronically compromised. This mechanistic segregation explains why acute synaptic blockade (shi^ts) disrupts feeding but not mating—shi^ts impair SCV recycling but spares DCV exocytosis. Conversely, chronic depletion of SIFa, ple, or Vglut compromises mating by directly or indirectly disabling neuropeptide signaling. These findings highlight a functional division of labor within SIFa neurons, where SCVs drive rapid behaviors (e.g., feeding) and DCVs mediate sustained neuromodulation (e.g., mating). Such multiplexed signaling enables a single neuronal population to coordinate diverse physiological outputs without behavioral interference, underscoring the adaptability of peptidergic circuits in encoding context-dependent responses.

Our study shows that SIFa neurons function as a central hub for integrating various internal signals and translating them into specific behavioral outputs, potentially offering novel insights into the orchestration of complex behaviors in other organisms. In short, this study demonstrates that both input signals mediated by neuropeptides and output signals transmitted through SIFa-SIFaR communication or diverse neurotransmitters can modulate a range of complex behaviors related to energy homeostasis, including sleep, feeding, courtship, and interval timing (summarized diagram in Fig 6N).

We discovered that sNPF functions as a crucial controller of interval timing behaviors through its receptor sNPF-R, which is expressed in SIFa neurons (as shown in Figs 4 and S6). In addition to its involvement in interval timing behavior, sNPF also contributes to the regulation of sleep and feeding in *D. melanogaster* [68,69,72]. The activation of SIFa- and sNPF-expressing neurons using optogenetics has been observed to induce sleep in fruit flies, but the mechanisms involved are quite distinct [70,46]. However, the exact functions of these neurons in regulating sleep are not yet completely understood. We demonstrated the cooperative interaction between the SIFa and sNPF systems through the establishment of a feedback loop, facilitated by the relay of neuropeptides and their subsequent binding to their respective receptors in specific neural circuits. Our studies revealed that SIFa neurons, situated in the brain, transmit signals related to internal states to sNPF neurons that express SIFaR in the VNC. This conclusion is substantiated by our discovery that SIFa establishes robust synaptic connections with sNPF-R neurons in the LAL, a brain region that crucially regulates behavior by linking the brain and thoracic motor center with a group of descending neurons. Further investigations into the downstream targets and modulatory processes of SIFa signaling offer promise for explaining its full contribution to behavioral regulation.

Our results provide direct evidence that sNPF neurons regulate SIFa neuronal activity, supporting the hypothesis that sNPF signaling mediates communication between hunger and satiety signals and SIFa neurons. Specifically, ATP-induced activation of sNPF neurons led to a significant increase in calcium activity in SIFa neurons, while no such activation occurred in the control condition. This finding highlights the functional connection between sNPF and SIFa neurons, suggesting that sNPF inputs are critical for the generation of SIFa-mediated internal states. The activation of SIFa neurons via sNPF inputs likely plays a pivotal role in interval timing behaviors, as demonstrated by our behavioral and genetic manipulations. These results complement previous findings that SIFa-SIFaR and sNPF-sNPF-R signaling are mutually dependent pathways for regulating timing-related behaviors. Additionally, the robust activation of SIFa neurons in response to sNPF stimulation underscores the importance of this neuropeptidergic interaction in modulating internal states.

The neuropeptides that communicate with SIFa neurons exert distinct influences on interval timing behaviors, highlighting the complexity of their regulatory mechanisms. The SIFa neurons receive inputs from many peptidergic pathways, including Crz, dilp2, Dsk, sNPF, MIP, and hugin [44,49,94]. Future studies exploring how these diverse inputs orchestrate physiological and behavioral outcomes will provide valuable insights into the context-dependent modulation of behaviors regulated by SIFa neurons.

Finally, the relationship between SIFaR and gonadotropin inhibitory hormone receptors (GnIHR) [95] highlights an intriguing evolutionary connection, as both are believed to have descended from a common ancestor [96,97]. This study expands on previous findings by Martelli and colleagues, demonstrating that SIFa not only regulates homeostatic

behaviors but also plays a significant role in reproductive behavior [44]. GnIHR regulates food intake and reproductive behavior in opposing directions, thereby prioritizing feeding behavior over other behavioral tasks during times of metabolic need [98]. The evolution of these behavioral control mechanisms suggests a complex interplay between neuropeptides that modulate both physiological states and reproductive strategies. As SIFa influences various behaviors, including feeding and sexual activity, it may be integral to understanding how organisms adapt their reproductive strategies in response to environmental and internal cues. This integration of behavioral modulation underscores the evolutionary significance of SIFa signaling in coordinating essential life functions in *Drosophila melanogaster* and potentially other species, revealing pathways through which neuropeptides can shape behavior across different contexts.

## Materials and methods

### Fly stocks and husbandry

*Drosophila Melanogaster* were raised on cornmeal-yeast medium at similar densities to yield adults with similar body sizes. Flies were kept in 12 h light: 12 h dark cycles (LD) at 25 °C (ZT 0 is the beginning of the light phase, ZT12 beginning of the dark phase) except for some experimental manipulation. For temperature-controlled experiments, the flies were initially crossed and maintained at a constant temperature of 22 °C within an incubator. The temperature shift was initiated post-eclosion. Once the flies had emerged, they were transferred to an incubator set at an elevated temperature of 29 °C for a defined period, after which the experimental protocols were carried out. Wild-type flies were *Canton-S*(CS).

Following lines used in this study, *Canton-S* (#64349), *w1118* (#3605), *Df(1)Exel6234* (#7708), *SIFa²ᴬ⁻ᴳᴬᴸ⁴* (#84690), *ple-RNAi* (#25796), *Trhn-RNAi* (#25842), *ChAT-RNAi* (#25856), *Gad1-RNAi* (#28079), *VGlut-RNAi* (#27538), *hdc-RNAi* (#30489), *SIFa-RNAi* (#60484), *sNPF-RNAi* (#25867), *sNPF-R-RNAi* (#27507), *SIFaR-RNAi* (#34947), *SIFaR²⁴ᶠ⁰⁶-GAL4* (#49087), *sNPF-R⁶⁴ᴮ¹¹-GAL4* (#41288), *sNPF-R⁶⁴ᴴ⁰⁹-GAL4* (#46547), *CCKLR17D1-RNAi* (#27494), *CCKLR17D3-RNAi* (#28333), *Dimm-RANi* (#38530), *UAS>stop>mCD8GFP* (#30125), *fruᶠᴸᴾ* (#66870), *UAS-KCNJ2* (#6596), *UAS-TNT* (#28838), *UAS-NaChBac-EGFP* (#9466), *SIFa[1]* (#80696), *SIFa[2]* (#80697), *UAS-myr-GFP* (#32197), *UAS-CD4tdGFP* (#35839), *UAS-Redstinger* (#32222), *UAS-mCD8RFP, lexAop-mCD8GFP* (#32229), *lexAop-nSyb-spGFP1-10, UAS-CD4-spGFP11* (#64315), *UAS-mCD8-RFP* (#61679), *UAS-myrGFP.QUAS-mtdTomato-3xHA, trans-Tango* (#77124), *elavᶜ¹⁵⁵; UAS-Dicer* (#25750), *UAS-Denmark, UAS-syt.eGFP* (#33065), *UAS-mCD8RFP, lexAop-mCD8GFP* (#32229), *UAS-Dop1R1-Tango* (#68234), *UAS-Dop2R-RNAi* (#26001), *UAS-Dop1R2-RNAi* (#51423), *UAS-DopEcR-RNAi* (#93710) were obtained from the Bloomington *Drosophila* Stock Center at Indiana University. The following lines, *CrzR-RNAi* (#108506), *PK2-R2-RNAi* (#44871), and *CCHa1-R-RNAi* (#103055). *InR-RNAi* (#991), *UAS-TrpA1* (#61504), *UAS-ShiBirets* (#66600), *UAS-dsxᶠ* (#44223), *UAS-traᶠ* (#4590), and *UAS-sxl* (#55769) were obtained from the Vienna *Drosophila* Resource Center. The following lines, *SIFa²ᴬ⁻ˡᵉˣᴬ* (#FBA00106), *SIFaR²ᴬ-GAL4* (#FBA00102), *UAS-t-SIFa-SEC, UAS-t-SIFa-ML, SIFaᴾᵀ-lexA* (see Materials and methods: Generation of transgenic flies and SIFa t-Peptide (t-SIFa) generation). The *GAL4ˢᴵᶠᵃ.ᴾᵀ* was a gift from Jan A. Veenstra. *UAS-OrkΔC* were obtained from Nitabach MN.

To reduce the variation from genetic background, all flies were backcrossed for at least 3 generations to CS strain. For the generation of outcrosses, all GAL4, UAS, and RNAi lines employed as the virgin female stock were backcrossed to the *CS* genetic background for a minimum of ten generations. Notably, the majority of these lines, which were utilized for LMD assays, have been maintained in a *CS* backcrossed state for long-term generations subsequent to the initial outcrossing process, exceeding ten backcrosses. Based on our experimental observations, the genetic background of primary significance is that of the X chromosome inherited from the female parent. Consequently, we consistently utilized these fully outcrossed females as virgins for the execution of experiments pertaining to LMD and SMD behaviors. Contrary to the influence on LMD behaviors, we have previously demonstrated that the genetic background exerts negligible influence on SMD behaviors, as reported in our prior publication [34]. The mutants and transgenic lines utilized in this study have been previously characterized, with the exception of the novel transgenic strains that we generated and describe herein.

## Mating duration assay

The MD assay in this study has been reported [34,29,28]. To enhance the efficiency of the MD assay, we utilized the *Df (1) Exel6234* (DF hereafter) genetic modified fly line in this study, which harbors a deletion of a specific genomic region that includes the sex peptide receptor (SPR) [99,100]. Previous studies have demonstrated that virgin females of this line exhibit increased receptivity to males [100]. We conducted a comparative analysis between the virgin females of this line and the CS virgin females and found that both groups induced SMD. Consequently, we have elected to employ virgin females from this modified line in all subsequent studies. For naïve males, 40 males from the same strain were placed into a vial with food for 5 days. For single-reared males, males of the same strain were collected individually and placed into vials with food for 5 days. For experienced males, 40 males from the same strain were placed into a vial with food for 4 days then 80 DF virgin females were introduced into vials for last 1 day before assay. Forty DF virgin females were collected from bottles and placed into a vial for 5 days. These females provide both sexually experienced partners and mating partners for MD assays. On the fifth day after eclosion, males of the appropriate strain and DF virgin females were mildly anaesthetized by $CO_2$. After placing a single female into the mating chamber, we inserted a transparent film and then placed a single male on the other side of the film in each chamber. After allowing for 1 h of recovery in the mating chamber in 25 °C incubators, we removed the transparent film and recorded the mating activities. Only those males that succeeded to mate within 1 h were included for analyses. Initiation and completion of copulation were recorded with an accuracy of 10 s, and total MD was calculated for each couple. All assays were performed from noon to 4 pm. Part of genetic controls with *GAL4*/+ or *UAS*/+ lines were omitted from supplementary figures, as prior data confirm their consistent exhibition of normal LMD and SMD behaviors [34,29,28,101,102]. Hence, genetic controls for LMD and SMD behaviors were incorporated exclusively when assessing novel fly strains that had not previously been examined. In essence, internal controls were predominantly employed in the experiments, as LMD and SMD behaviors exhibit enhanced statistical significance when internally controlled. Within the LMD assay, both group and single conditions function reciprocally as internal controls. A significant distinction between the naïve and single conditions implies that the experimental manipulation does not affect LMD. Conversely, the lack of a significant discrepancy suggests that the manipulation does influence LMD. In the context of SMD experiments, the naïve condition (equivalent to the group condition in the LMD assay) and sexually experienced males act as mutual internal controls for one another. A statistically significant divergence between naïve and experienced males indicates that the experimental procedure does not alter SMD. Conversely, the absence of a statistically significant difference suggests that the manipulation does impact SMD. Hence, we incorporated supplementary genetic control experiments solely if they deemed indispensable for testing. All assays were performed from noon to 4 PM. We conducted blinded studies for every test [103,104].

## Quantitative analysis of fluorescence intensity

To ascertain calcium levels and synaptic intensity from microscopic images, we dissected and imaged five-day-old flies of various social conditions and genotypes under uniform conditions. For group-reared (naïve) flies, the flies were reared in group condition and dissect right after 5 days of rearing without any further action. For single-reared flies, the flies were reared in single condition and dissect at the same time as group-reared flies right after 5 days of rearing without any further action. For sexual experienced flies, the flies were reared in group condition after 4 days of rearing and will be given virgins to give them sexual experience for 1 day, those flies will also be dissected at the same time as group and single-reared flies after one day. The GFP signal in the brains and VNCs was amplified through immunostaining with chicken anti-GFP, rabbit anti-DsRed, and mouse anti-nc82 primary antibodies. Image analysis was conducted using ImageJ software. For the quantification of fluorescence intensities, an investigator, blinded to the fly's genotype, thresholded the sum of all pixel intensities within a sub-stack to optimize the signal-to-noise ratio, following established methods [105]. The total fluorescent area or region of interest (ROI) was then quantified using ImageJ, as previously reported. For CaLexA or TRIC signal quantification, we adhered to protocols detailed by Kayser and colleagues [106], which involve

measuring the ROI's GFP-labeled area by summing pixel values across the image stack. This method assumes that changes in the GFP-labeled area and intensity are indicative of alterations in the CaLexA and TRIC signal, reflecting synaptic activity. ROI intensities were background-corrected by measuring and subtracting the fluorescent intensity from a nonspecific adjacent area, as per Kayser and colleagues [106]. For normalization, nc82 fluorescence is utilized for CaLexA, while RFP signal is employed for TRIC experiments, as the RFP signal from the TRIC reporter is independent of calcium signaling [80]. For the analysis of GRASP or tGRASP signals, a sub-stack encompassing all synaptic puncta was thresholded by a genotype-blinded investigator to achieve the optimal signal-to-noise ratio. The fluorescence area or ROI for each region was quantified using ImageJ, employing a similar approach to that used for CaLexA or TRIC quantification [105]. 'Norm. GFP Int.' refers to the normalized GFP intensity relative to the RFP signal.

### Image display and thresholding for figure preparation

To prepare representative images for figures, grayscale channels were used for single fluorophore display to improve clarity and avoid color-based misinterpretation. For overlays involving multiple reporters (e.g., GFP, RFP, and nc82), distinct non-overlapping pseudocolors were applied, with channel identities labeled below each panel. Thresholding for all displayed images was performed conservatively and consistently across genotypes and conditions, with the primary goal of preserving visible biological signals while avoiding oversaturation. For quantified data, identical threshold settings were applied across conditions. All brightness and contrast adjustments were applied uniformly to the entire image and were restricted to linear modifications for visual clarity only. No nonlinear enhancements or selective adjustments were applied. These standardized procedures ensured both accurate visualization and reproducible quantification. Raw image data are available upon request.

### Generation of transgenic flies

To generate the *SIFa^PT-lexA* driver, the putative promotor sequence of the gene was amplified by PCR using wild-type genomic DNA as a template with the following primers GCCAATTGGCTGAATCTCCTGACCCTCA and GCAGATCTCTTGCAGTTTTCGGTGAGC as mentioned before [107]. The amplified DNA fragment (1482 base pairs located immediately upstream of the *SIFa* coding sequence) was inserted into the E2 Enhancer-lexA vector. This vector, supplied by Qidong Fungene Biotechnology Co., (http://www.fungene.tech/), is a derivative of the pBPLexA::p65Uw vector (available at https://www.addgene.org/26231). The insertion was achieved by digesting the fragment and the vector with EcoRI and XbaI restriction enzymes to create compatible sticky ends. The genetic construct was inserted into the attp2 site on chromosome III to generate transgenic flies using established techniques, a service conducted by Qidong Fungene Biotechnology Co., Ltd.

### SIFa t-Peptide (t-SIFa) generation

To generate the *UAS-t-SIFa-ML* and *UAS-t-SIFa-SEC* driver in Fig 1E–1H, t-peptide cDNAs were chemically synthesized with optimal *Drosophila* codon usage and with an optimal *Drosophila* Kozak translation initiation site upstream of the start methionine (CAAA) [108]. Encoded t-peptide is as follows: *t-SIFa-ML*, MSALLILALVGAAVAAYRKPPFNGSIFGNEQKLI-SEEDLGNGAGFATPVTLALVPALLATFWSLL. These cDNAs were cloned into the pUAS-attB vector. For generation of transgenic *Drosophila*, vectors were injected into the embryos of flies. The genetic construct was inserted into the attp40 site on chromosome II to generate transgenic flies using established techniques, a service conducted by Qidong Fungene Biotechnology Co., Ltd. (http://www.fungene.tech/).

### Antibody preparation

To generate the SIFa-antibody in S1C Fig, the peptide sequence RDKKTLKIKMKLFI of SIFa was synthesized and conjugated to Keyhole Limpet Hemocyanin (KLH). The efficiency of the conjugation was confirmed by SDS-PAGE analysis. Healthy rabbits were immunized via multiple subcutaneous injections with the peptide-KLH conjugate mixed with Freund's

adjuvant. The immunization schedule included a primary immunization and two booster injections, each separated by 3 weeks. Two weeks after the final booster, serum was collected from the rabbits via cardiac puncture. The serum was purified using affinity chromatography, and the antibody titer and specificity were determined by ELISA. Antibody synthesis was performed by Wuhan Vapol Bioscience Company (http://www.vapolbio.com).

### Courtship assay

Courtship assay was performed as previously described [109], under normal light conditions in circular courtship arenas 11 mm in diameter, from noon to 4 PM Courtship latency is the time between female introduction and the first obvious male courtship behavior, such as orientation coupled with wing extensions. Once courtship began, courtship index was calculated as the fraction of time a male spent in any courtship-related activity during a 10 min period or until mating occurred.

### Immunostaining

The dissection and immunostaining protocols for the experiments are described elsewhere [28]. After 5 days of eclosion, the *Drosophila* brain was taken from adult flies and fixed in 4% formaldehyde at room temperature for 30 min. The sample was washed three times (5 min each) in 1% PBT and then blocked in 5% normal goat serum for 30 min. The sample next be incubated overnight at 4 °C with primary antibodies in 1% PBT, followed by the addition of fluorophore-conjugated secondary antibodies for one hour at room temperature. The brain was mounted on plates with an antifade mounting solution (Solarbio) for imaging purposes.

Samples were imaged with Zeiss LSM880. Antibodies were used at the following dilutions: Chicken anti-GFP (1:500, Invitrogen), mouse anti-nc82 (1:50, DSHB), rabbit anti-DsRed (1:500, Rockland Immunochemicals), Alexa-488 donkey anti-chicken (1:200, Jackson ImmunoResearch), Alexa-555 goat anti-rabbit (1:200, Invitrogen), Alexa-647 goat anti-mouse (1:200, Jackson ImmunoResearch).

### RNA extraction and cDNA synthesis

RNA was extracted from 50 preparations of 5-day-old CS males using the RNA isolation kit (Vazyme), following the manufacturer's protocol. And first-strand cDNA was synthesized from 1 µg of RNA template with random primers using SPARK script II RT plus kit (SparkJade).

### Quantitative RT-PCR

The expression levels of *SIFa* in naïve, exp, and single conditions were analyzed by quantitative real-time RT-PCR with SYBR Green qPCR MasterMix kit (Selleckchem). The primers of RT-PCR are F:5′-aagcaggagagcgagttcag-3′; R: 5′-ttcgccttgttttgtcacag-3′ [103]. qPCR reactions were performed in triplicate, and the specificity of each reaction was evaluated by dissociation curve analysis. Each experiment was replicated three times. PCR results were recorded as threshold cycle numbers (Ct). The fold change in the target gene expression, normalized to the expression of internal control gene (GAPDH) and relative to the expression at time point 0, was calculated using the $2^{-\Delta\Delta CT}$ method as previously described [104]. The results are presented as the mean ± SD of three independent experiments.

### Colocalization analysis

Before the colocalization analysis, an investigator, blinded to the fly's genotype, threshed the sum of all pixel intensities within a sub-stack to optimize the signal-to-noise ratio, following established methods [105]. To perform colocalization analysis of multi-color fluorescence microscopy images in this study, we employed ImageJ software [110]. In brief, we merged image channels to form a composite with accurate color representation and applied a threshold to isolate yellow

pixels, signifying colocalization. The measured "area" values represented the colocalization zones between fluorophores. To determine the colocalization percentage relative to the total area of interest (e.g., GFP or RFP), we adjusted thresholds to capture the full fluorophore areas and remeasured to obtain total areas. The colocalization efficiency was calculated by dividing the colocalized area by the total fluorophore area. All samples were imaged uniformly.

### Locomotion assay

To detect and quantify the activity of flies, we used the Fly Trajectory Dynamics Tracking (FlyTrDT) software. This is an open-source, custom-written Python program that utilizes the free OpenCV machine vision library and the Python Qt library. The FlyTrDT software simultaneously records the trajectory information of each fly and calculates various indicators of the group at a certain period. For each frame acquired, the moving fly is segmented using the binarization function from the OpenCV library. Subsequently, a Gaussian blur and morphological closing and opening operations were performed on the extracted foreground pixels to consolidate detected features and reducing false positives and negatives. Finally, the extraction of fly outlines was achieved using the contour detection algorithm in the OpenCV library.

### Sleep assay of single fly

Adult male flies (aged 3–7 days) were loaded into wells of white 96-well Microfluor 2 plates (Fishier) containing 300 μl food (5% sucrose and 1% agar). The fly boxes were placed in an incubator to control temperature precisely. A webcam tracked the locomotion of flies at 10 s intervals, and the data were used for sleep analysis [111]. All experiments were replicated at least three times.

### Sleep assay of group reared flies

To investigate *Drosophila* circadian rhythms, we utilized the DAM system [112], which records infrared beam breaks as flies move through glass tubes. Each DAM monitor tracked 32 individual flies, with experiments scaling to hundreds. Flies were housed in tubes with food for extended monitoring, and drugs were added to modulate behavior. Monitors were placed in incubators controlling temperature, humidity, and light to study environmental responses and transgene expression. We developed ShinyR-DAM, a free, open-source, cloud-based application for rapid data analysis, producing customizable plots and CSV files. ShinyR-DAM streamlines the analysis of LD and DD conditions, eliminating complex installations and manual dead fly detection. It leverages the R programming language's Shiny framework, making it accessible and customizable for researchers.

### Calcium imaging

The entire brain was dissected out in a plate filled with *Drosophila* Adult Hemolymph-Like Saline (AHL) buffer [108 mM NaCl, 5 mM KCl, 4 mM NaHCO$_3$, 1 mM NaH$_2$PO$_4$, 15 mM ribose, mM Hepes (pH 7.5); 300 mosM], CaCl$_2$ (2 mM), and MgCl$_2$ (8.2 mM) were added to the AHL before use to ensure the flies' neurons remained in an active state throughout the experiment [113,114]. Calcium imaging was performed using an fluorescent microscope with a 10X objective. The Ca$^{2+}$ indicator GCaMP6 was used to measure the Ca$^{2+}$ signal. GCaMP6 was excited with 488 nm light and the fluorescent signals were collected at 5 Hz. ATP solution was added into the chamber to a final ATP concentration of 10 mM[119]. Regions of interest (ROIs) were manually selected from the cell body with ImageJ. Fluorescent change was calculated as $(F_{peak} - F_0)/F_0$, where $F_0$ was calculated from the average intensity of 50 frames of background-subtracted baseline fluorescence before optogenetic stimulation, and $F_{peak}$ corresponds to the highest fluorescence after stimulation. To quantify neuronal activity in the brain region, the brains of male flies with the appropriate genotype were dissected and secured in a plastic dish. All subsequent steps were conducted as previously described.

## Fluorescence imaging (rDA1m)

The entire brain was dissected out in a plate filled with *Drosophila* Adult Hemolymph-Like Saline (AHL) buffer [108 mM NaCl, 5 mM KCl, 4 mM NaHCO$_3$, 1 mM NaH$_2$PO$_4$, 15 mM ribose, mM Hepes (pH 7.5); 300 mosM], CaCl$_2$ (2 mM), and MgCl$_2$ (8.2 mM) were added to the AHL before use to ensure the flies' neurons remained in an active state throughout the experiment [113,114]. Fluorescence imaging was performed using an fluorescent microscope with a 20× objective. The Dopamine sensor rDA1m was used to measure the dopamine signal. rDA1m was excited with 555 nm light and the fluorescent signals were collected at 5 Hz. ATP solution was added into the chamber to a final ATP concentration of 10 mM [115]. Regions of interest (ROIs) were manually selected from the cell body with ImageJ. Fluorescent change was calculated as $(F_{peak} - F_0)/F_0$, where $F_0$ was calculated from the average intensity of 50 frames of background-subtracted baseline fluorescence before optogenetic stimulation, and $F_{peak}$ corresponds to the highest fluorescence after stimulation.

## Blue dye assay

For colorimetric food intake assay, flies were starved in PBS-containing vials for 2 h and fed for 15 min in vials containing 0.05% FD&C Blue dye, 7% sucrose and 5% yeast. The flies were frozen, homogenized in PBS, and centrifuged twice for 25 min each. The supernatant was measured at 625 nm. Each experiment consisted of 20 flies and the assay was repeated three times [83].

## Ex-Q assay

Two flies were placed in a tube with 5% sucrose, 5% yeast, and 1% agar with 0.5% erioglaucine disodium blue dye. After 24 h, flies were homogenized in tubes with its excreta and centrifuged for 15 min in 13,000 rpm. The supernatant was measured at 630 nm. For TrpA1 and shi$^{ts}$ thermogenetic experiments, flies were maintained and starved in 20 °C and transferred to 30 °C 30 min before testing.

## Single-nucleus RNA-sequencing analyses

The snRNAseq dataset analyzed in this paper is published in [116] and available at the Nextflow pipelines (VSN, https://github.com/vib-singlecell-nf), the availability of raw and processed datasets for users to explore, and the development of a crowd-annotation platform with voting, comments, and references through SCope (https://flycellatlas.org/scope), linked to an online analysis platform in ASAP (https://asap.epfl.ch/fca). For the generation of the tSNE plots, we utilized the Fly SCope website (https://scope.aertslab.org/#/FlyCellAtlas/*/welcome). Within the session interface, we selected the appropriate tissues and configured the parameters as follows: 'Log transform' enabled, 'CPM normalize' enabled, 'Expression-based plotting' enabled, 'Show labels' enabled, 'Dissociate viewers' enabled, and both 'Point size' and 'Point alpha level' set to maximum. For all tissues, we referred to the individual tissue sessions within the '10X Cross-tissue' RNAseq dataset. Each tSNE visualization depicts the coexpression patterns of genes, with each color corresponding to the genes listed on the left, right, and bottom of the plot. The tissue name, as referenced on the Fly SCope website is indicated in the upper left corner of the tSNE plot. Dashed lines denote the significant overlap of cell populations annotated by the respective genes. Coexpression between genes or annotated tissues is visually represented by differentially colored cell populations. For instance, yellow cells indicate the coexpression of a gene (or annotated tissue) with red color and another gene (or annotated tissue) with green color. Cyan cells signify coexpression between green and blue, purple cells for red and blue, and white cells for the coexpression of all three colors (red, green, and blue). Consistency in the tSNE plot visualization is preserved across all figures.

## Statistical tests

Statistical analysis of MD assay was described previously [28,29,34]. More than 50 males (naïve, experienced, and single) were used for MD assay. Our experience suggests that the relative MD differences between naïve and experienced

condition and singly reared are always consistent; however, both absolute values and the magnitude of the difference in each strain can vary. So, we always include internal controls for each treatment as suggested by previous studies. Therefore, statistical comparisons were made between groups that were naïvely reared, sexually experienced and singly reared within each experiment. As MD of males showed normal distribution (Kolmogorov–Smirnov tests, $p > 0.05$), we used two-sided Student $t$ tests. The mean ± standard error (s.e.m) (**** = $p < 0.0001$, *** = $p < 0.001$, ** = $p < 0.01$, * = $p < 0.05$). All analysis was done in GraphPad (Prism). Individual tests and significance are detailed in Figure legends.

Besides traditional $t$ test for statistical analysis, we added estimation statistics for all MD assays and two group comparing graphs. In short, 'estimation statistics' is a simple framework that—while avoiding the pitfalls of significance testing—uses familiar statistical concepts: means, mean differences, and error bars. More importantly, it focuses on the effect size of one's experiment/intervention, as opposed to significance testing. In comparison to typical NHST plots, estimation graphics have the following five significant advantages such as (1) avoid false dichotomy, (2) display all observed values (3) visualize estimate precision (4) show mean difference distribution. And most importantly (5) by focusing attention on an effect size, the difference diagram encourages quantitative reasoning about the system under study. Thus, we conducted a reanalysis of all of our two group data sets using both standard t tests and estimate statistics. In 2019, the Society for Neuroscience journal eNeuro instituted a policy recommending the use of estimation graphics as the preferred method for data presentation.

## Supporting information

**S1 Movie. 3D immunofluorescent visualization of SIFa neurons in the _Drosophila_ brain.** Flies expressing _GAL-4$^{SIFa.PT}$_ together with _UAS-RedStinger_ were immunostained with anti-elav (green), anti-repo (blue) and anti-SIFa (red) antibodies.
(AVI)

**S1 Fig. SIFa-expressing neurons are not linked to sexual dimorphism. (A)** Flies expressing _GAL4$^{SIFa.PT}$_/+(top) or _elav-GAL80; GAL4$^{SIFa.PT}$_ drivers(bottom) together with _UAS-RedStinger_ were immunostained with anti-SIFa (red) antibodies. Red arrowheads indicate SIFa region labeled by anti-SIFa antibodies. The right panels are presented as a gray scale to clearly show the axon expression patterns of _SIFa_ neurons in the adult brain labeled by _GAL4_ drivers. Scale bars represent 10 μm. **(B)** Flies expressing _GAL4$^{SIFa.PT}$_ together with _UAS-RedStinger_ were immunostained with anti-elav (green), anti-repo (blue) and anti-SIFa(red) antibodies. Arrowheads: cell body locations. Scale bars represent 10 μm. **(C)** Flies expressing _SIFa-RNAi_ with _GAL4$^{SIFa.PT}$_ or _w$^{1118}$_, _elav$^{c155}$_ and _repo-GAL4_ drivers (from top to bottom) were immunostained with anti-SIFa (green) antibodies. The right panels are presented as a gray scale to clearly show the axon expression patterns of SIFa neurons in the adult brain labeled by _GAL4_ drivers. Scale bars represent 10 μm. **(D)** qRT-PCR results show that SIFa expression level in group (gray), single (blue) and exp (red) conditions. The y-axis depicts the relative expression level of SIFa, normalized to the CT value of the GAPDH gene. "RGX" denotes relative gene expression. See the Materials and methods for a detailed description of the Quantitative RT-PCR used in this study. Statistical significance determined by one-way ANOVA followed by Tukey's comparisons test. **$p < 0.01$ (one-way ANOVA, $F = 26.09$, $R^2 = 0.8969$; Tukey's post-hoc). The ns represents non-significant differences. Sample sizes ($n$) are indicated in the figure panels. **(E)** _SIFa_ neurons are categorized into two subpopulations, the anterior-dorsal _SIFa_ neurons (SIFa$^{DA}$) and the posterior-ventral _SIFa_ neurons (SIFa$^{VP}$), based on their anatomical positioning and putative functional roles. D: dorsal; V: ventral; A: anterior; P: posterior. Scale bars represent 10 μm. **(F)** The brains of CS _Drosophila_ (left panel) and the VNC (right panel) were subjected to immunostained with anti-SIFa antibodies (red channel) under naïve, single, and exp conditions. Scale bars represent 100 μm in brain and 10 μm. **(G, H)** Quantification of SIFa signals in S1F Fig (two-tailed unpaired $t$ test). In all plots and statistical tests. Data are presented as mean ± s.e.m. ns = not significant ($p > 0.05$), *$p < 0.05$, **$p < 0.01$, ***$p < 0.001$, ****$p < 0.0001$. **(I)** The threshold of GFP fluorescence in male and female fly brain was marked by threshold

function of ImageJ. **(J)** Quantification of male and female brain regions covered by the SIFa cell membrane (two-tailed unpaired $t$ test). In all plots and statistical tests. Data are presented as mean±s.e.m. ns=not significant ($p > 0.05$), *$p < 0.05$, **$p < 0.01$, ***$p < 0.001$, ****$p < 0.0001$. **(K)** Expression pattern of SIFa in Virtual Fly Brain (VFB). **(L)** Flies expressing $GAL4^{SIFa.PT}$ drivers together with $UAS>stop>mCD8GFP$; $fru^{FLP}$ was immunostained with anti-GFP (green), anti-RFP (red), and nc82 (blue) antibodies. Scale bars represent 100 μm. Underlying data for all graphs can be found in file S1 Data.
(TIF)

**S2 Fig. SIFa neurons display neuronal processes which cover the entire brain, regardless of sexes. (A–F)** MD assays for $GAL4^{SIFa.PT}$ drivers mediated expression of female form of doublesex ($UAS-dsx^F$), transformer ($UAS-tra^F$), or sex lethal ($UAS-sxl$). **(G)** Flies expressing $GAL4^{SIFa.PT}$ drivers together with $UAS-tdTomato$ were immunostained with anti-RFP (yellow), and nc82 (blue) antibodies. Areas outlined by boxes are enlarged in the bottom panel, respectively. Arrowheads: cell body locations. Scale bars represent 100 μm. **(H, I)** Quantification of RFP fluorescence in the male (left) and female (right) fly brain expressing $GAL4^{SIFa.PT}$ driver together with $UAS-tdTomato$. (H) The threshold of RFP fluorescence (upper panel), nc82 (bottom panel) in male and female fly brain was marked by threshold function of ImageJ. (I) Quantification of RFP fluorescence in male (blue) and female (pink) brain (two-tailed unpaired $t$ test). In all plots and statistical tests. Data are presented as mean±s.e.m. ns=not significant ($p > 0.05$), *$p < 0.05$, **$p < 0.01$, ***$p < 0.001$, ****$p < 0.0001$. The same symbols for statistical significance are used in all other figures. See the Materials and methods for a detailed description of the colocalization analysis used in this study. **(J, K)** Quantification of RFP fluorescence in brain and VNC of male fly (two-tailed unpaired $t$ test). In all plots and statistical tests. Data are presented as mean±s.e.m. ns=not significant ($p > 0.05$), *$p < 0.05$, **$p < 0.01$, ***$p < 0.001$, ****$p < 0.0001$. **(L, M)** Quantification of RFP fluorescence of CB and OL area in male (L, M) and female **(N, O)** (two-tailed unpaired $t$ test). In all plots and statistical tests. Data are presented as mean±s.e.m. ns=not significant ($p > 0.05$), *$p < 0.05$, **$p < 0.01$, ***$p < 0.001$, ****$p < 0.0001$. **(P)** Larva expressing $GAL4^{SIFa.PT}$ drivers together with $UAS-tdTomato$ were immunostained with anti-RFP (yellow), and nc82 (blue) antibodies. Scale bars represent 100 μm. **(Q)** Quantification of the RFP fluorescence in third instar larvae and adult (two-tailed unpaired $t$ test). In all plots and statistical tests. Data are presented as mean±s.e.m. ns=not significant ($p > 0.05$), *$p < 0.05$, **$p < 0.01$, ***$p < 0.001$, ****$p < 0.0001$. **(R)** The threshold of RFP fluorescence (top panel), nc82 (bottom panel) in male larva was marked by threshold function of ImageJ. Underlying data for all graphs can be found in file S1 Data.
(TIF)

**S3 Fig. Presynaptic terminals are arborized more than dendrites of *SIFa* neurons. (A and K)** Distribution of dendrites and presynaptic terminals of neurons labeled via $GAL4^{SIFa.PT}$ in the brain and VNC. Flies expressing $GAL4^{SIFa.PT}$ together with $UAS-Denmark$, $UAS-syt.eGFP$ were immunostained with anti-GFP (green), anti-DsRed (red), and nc82 (blue) antibodies. Areas outlined by boxes are enlarged in bottom panel. Scale bars represent 100 μm in A and 50 μm in K. **(B–J)** Colocalization analysis of dendritic and presynaptic terminals of neurons labeled via $GAL4^{SIFa.PT}$ in the brain (B–D), CB (E–G), and OL (H–J) (two-tailed unpaired $t$ test). In all plots and statistical tests. Data are presented as mean±s.e.m. ns=not significant ($p > 0.05$), *$p < 0.05$, **$p < 0.01$, ***$p < 0.001$, ****$p < 0.0001$. Sample sizes ($n$) are indicated in the figure panels. See the Materials and methods for a detailed description of the colocalization analysis used in this study. **(L–N)** Quantification of dendritic and presynaptic terminals of neurons labeled via $GAL4^{SIFa.PT}$ in the brain and VNC (two-tailed unpaired $t$ test). In all plots and statistical tests. Data are presented as mean±s.e.m. ns=not significant ($p > 0.05$), *$p < 0.05$, **$p < 0.01$, ***$p < 0.001$, ****$p < 0.0001$. Sample sizes ($n$) are indicated in the figure panels. **(O)** Schematic shows that SIFa$^+$ cells form extensive synapses each other within the brain but not in the VNC. **(P)** Male flies expressing $SIFa^{2A-GAL4}$ together with $UAS-CD4tdGFP$ and $UAS-RedStinger$ were immunostained with anti-GFP (green), anti-RFP (red), and nc82 (blue) antibodies. Scale bars represent 100 μm. Underlying data for all graphs can be found in file S1 Data.
(TIF)

**S4 Fig. Various peptidergic inputs to *SIFa*. (A, B)** courtship index of control flies in group, single, and exp conditions. MD assays for *GAL4^SIFa.PT^* mediated knockdown *via empty-RNAi* (two-tailed unpaired *t* test). In all plots and statistical tests. Data are presented as mean ± s.e.m. ns = not significant ($p > 0.05$), *$p < 0.05$, **$p < 0.01$, ***$p < 0.001$, ****$p < 0.0001$. Sample sizes (*n*) are indicated in the figure panels. **(C–F)** Courtship index of flies for *GAL4*-mediated knockdown of *VGlut* (Glutamatergic), and *Ple* (Dopaminergic) via *VGlut-RNAi* and *ple-RNAi* using the *GAL4^SIFa.PT^* (two-tailed unpaired *t* test). In all plots and statistical tests. Data are presented as mean ± s.e.m. ns = not significant ($p > 0.05$), *$p < 0.05$, **$p < 0.01$, ***$p < 0.001$, ****$p < 0.0001$. Sample sizes (*n*) are indicated in the figure panels. **(G–L)** MD assays for *GAL4*-mediated knockdown of *GluRIIB*(G-H), *GluRIIE(I)*, *Nmdar2(J-K)* and *GluRIIC(L)* using *SIFaR^24F06^-GAL4* driver (two-tailed unpaired *t* test). In all plots and statistical tests. Data are presented as mean ± s.e.m. ns = not significant ($p > 0.05$), *$p < 0.05$, **$p < 0.01$, ***$p < 0.001$, ****$p < 0.0001$. Sample sizes (*n*) are indicated in the figure panels. **(M)** Control of the Fig 2P. Fly's brain after expression of the B2 recombinase in neurons of SIFa, Prior to excision of the STOP cassette smFLAG-vGlut is not expressed, above flies with UAS-mCD8RFP were immunostained with anti-DsRed (red), anti-FLAG (green), and nc82 (blue) antibodies. Scale bars represent 100 mm. Underlying data for all graphs can be found in file S1 Data. (TIF)

**S5 Fig. Dopamine receptor RNAi and neuronal expression analysis in SIFa neurons. (A–H)** MD assays of flies expressing the *SIFaR^24F06^* driver together with (A, B) *Dop1R1-RNAi*, (C, D) *Dop1R2-RNAi*, (E, F) *DopEcR-RNAi,* (G, H) *Dop2R-RNAi* (two-tailed unpaired *t* test). In all plots and statistical tests. Data are presented as mean ± s.e.m. ns = not significant ($p > 0.05$), *$p < 0.05$, **$p < 0.01$, ***$p < 0.001$, ****$p < 0.0001$. Sample sizes (*n*) are indicated in the figure panels. **(I)** Control for Fig 3A. Male flies expressing *SIFaR^24F06^* drivers together with *UAS-mCD8RFP* were imaged live under a fluorescent microscope with anti-GFP (green), anti-DsRed (red), and nc82 (blue) antibodies. Areas outlined by boxes are enlarged in the bottom panel. Scale bars represent 100 µm. **(J)** Flies expressing *LexA^SIFa.PT^* with *TH-C-GAL4* drivers together with *UAS-stinger*, *LexAop-tdTomato.nls* was immunostained with anti-GFP (green), anti-RFP (red), and nc82 (blue) antibodies. Scale bars represent 100 µm. **(K)** Flies expressing *LexA^2A-lexA^* with *TH-F-GAL4* drivers together with *UAS-stinger*, *LexAop-tdTomato.nls* was immunostained with anti-GFP (green), anti-RFP (red), and nc82 (blue) antibodies. Scale bars represent 100 µm. Underlying data for all graphs can be found in file S1 Data. (TIF)

**S6 Fig. SIFa and sNPF collaborate to regulate interval timing behaviors. (A–D)** MD assays for *GAL4*-mediated knockdown of *SIFaR* and *sNPF-R* via *SIFaR-RNAi* and *sNPF-R-RNAi* using the *sNPF-R^64H09^-GAL4* driver (A, B), and *SIFaR^24F06^-GAL4* driver (C, D) (two-tailed unpaired *t* test). In all plots and statistical tests. Data are presented as mean ± s.e.m. ns = not significant ($p > 0.05$), *$p < 0.05$, **$p < 0.01$, ***$p < 0.001$, ****$p < 0.0001$. Sample sizes (*n*) are indicated in the figure panels. **(E)** Male VNC of flies expressing *sNPF-R^64H09^-GAL4* and *SIFa^2A-lexA^* drivers together with *UAS-mCD8RFP* and *lexAop-mCD8GFP* was immunostained with anti-GFP (green), anti-DsRed (red), and anti-nc82 (blue) antibodies. The top two panels are presented as a gray scale to clearly show the membrane expression patterns of *SIFa^+^* neurons in the adult labeled by *sNPF-R^64H09^-GAL4* driver. Scale bars represent 100 µm. **(F, I)** Male brain and VNC of flies expressing *sNPF^2A-GAL4^* together with *UAS-mCD8RFP* was immunostained with anti-GFP (green), anti-DsRed (red), and anti-nc82 (blue) antibodies. Scale bars represent 100 µm. Areas outlined by boxes are enlarged in the bottom panel. Scale bars represent 50 µm. **(G)** Flies expressing *LexA^SIFa.PT^* and *sNPF-R^2A-GAL4^* drivers together with *UAS-stinger, LexAop-tdTomato.nls* was immunostained with anti-GFP (green), anti-RFP (red), and nc82 (blue) antibodies. Scale bars represent 100 µm. **(H)** Fly SCope single-cell RNA sequencing data of cells co-expressing SIFa together with sNPF-R. **(J)** GRASP assay for *SIFa^2A-lexA^* and *sNPF-R^64B11^-GAL4* in male brain (left three columns) and VNC (right three columns). Male flies expressing *SIFa^2A-lexA^*, *sNPF-R^64B11^-GAL4*, and *lexAop-nsyb-spGFP1-10, UAS-CD4-spGFP11* were dissected after 5 days of growth (mated male flies had 1 day of sexual experience with virgin females). The white dashed line highlights the GRASP signal. Scale bars represent 100 µm. **(K–M)** Quantification of synapses formed between *SIFa^2A-lexA^*

and *sNPF-R^64H09^-GAL4* in the brain. (K) The threshold of GFP fluorescence (left panel), nc82 (right panel) of naïve, single, and exp in male fly brain was marked by threshold function of ImageJ. (L, M) Quantification of synapses formed between *SIFa^2A-lexA^* and *sNPF-R^64H09^-GAL4* in the brain. Revealed by the GRASP system in naïve, single, and mated male flies. GFP fluorescence was normalized to that in nc82 (two-tailed unpaired *t* test). In all plots and statistical tests. Data are presented as mean±s.e.m. ns=not significant ($p > 0.05$), *$p < 0.05$, **$p < 0.01$, ***$p < 0.001$, ****$p < 0.0001$. See the Materials and methods for a detailed description of the Quantitative analysis of fluorescence intensity used in this study. Underlying data for all graphs can be found in file S1 Data.
(TIF)

**S7 Fig. Deactivating SIFa neurons may reduce food intake. (A–D)** 24-h food intake of males measured by EX-Q and Blue-Dye assays of male flies expressing *GAL4^SIFa.PT^* driver together with *TrpA1* on yeast-sugar medium of different temperatures (A, 29 °C; B, 22 °C; C, 29 °C; D, 22 °C) (two-tailed unpaired *t* test). In all plots and statistical tests. Data are presented as mean±s.e.m. ns=not significant ($p > 0.05$), *$p < 0.05$, **$p < 0.01$, ***$p < 0.001$, ****$p < 0.0001$. Sample sizes (*n*) are indicated in the figure panels. See the Materials and methods for a detailed description of the EX-Q and Blue-Dye assay used in this study. **(E–H)** Blue-Dye and EX-Q assay of flies expressing *GAL4^SIFa.PT^* driver together with *UAS-shi^ts^* in 22 and 29 °C (two-tailed unpaired *t* test). In all plots and statistical tests. Data are presented as mean±s.e.m. ns=not significant ($p > 0.05$), *$p < 0.05$, **$p < 0.01$, ***$p < 0.001$, ****$p < 0.0001$. Sample sizes (*n*) are indicated in the figure panels. **(I, J)** MD assays for *GAL4^SIFa.PT^* mediated the inactivation of synaptic transmission of SIFa neurons using *UAS-TNT* (two-tailed unpaired *t* test). In all plots and statistical tests. Data are presented as mean±s.e.m. ns=not significant ($p > 0.05$), *$p < 0.05$, **$p < 0.01$, ***$p < 0.001$, ****$p < 0.0001$. Sample sizes (*n*) are indicated in the figure panels. **(M-N, Q-R, U-V)** MD assays of *UAS-KCNJ2-eGFP* (M-N), *UAS-NachBac-eGFP* (Q-R), and *UAS-OrkΔC* (U-V) crossed with *GAL4^SIFa.PT^* (two-tailed unpaired *t* test). In all plots and statistical tests. Data are presented as mean±s.e.m. ns=not significant ($p > 0.05$), *$p < 0.05$, **$p < 0.01$, ***$p < 0.001$, ****$p < 0.0001$. Sample sizes (n) are indicated in the figure panels. **(K-L, O-P, S-T)** 24-h excreta quantification of males expressing *GAL4^SIFa.PT^* driver together with *UAS-TNT* (K), *UAS-KCNJ2* (M), and *UAS-NachBac* (Q) on yeast-sugar medium. Blue-Dye assay of male flies expressing *GAL4^SIFa.PT^* driver together with *UAS-TNT* (L), *UAS-KCNJ2* (P), and *UAS-NachBac* (T) (two-tailed unpaired *t* test). In all plots and statistical tests. Data are presented as mean±s.e.m. ns=not significant ($p > 0.05$), *$p < 0.05$, **$p < 0.01$, ***$p < 0.001$, ****$p < 0.0001$. Sample sizes (*n*) are indicated in the figure panels. **(W)** Schematic representation of the regulatory interactions between *SIFa* and *SIFaR*, in the context of modulating internal states. Underlying data for all graphs can be found in file S1 Data.
(TIF)

**S8 Fig. *SIFa* neurons project to important regions for control of reproductive behavior. (A)** Confocal image of male brain and VNC in which *GAL4^SIFa.PT^* was used to simultaneously drive expression of *trans-Tango*(magenta), visualized with RFP antibody staining, which identifies postsynaptic cells, and *myrGFP* (green), which labels the *GAL4^SIFa.PT^* cells. Areas outlined by boxes in top panels are enlarged in middle. The bottom panels are presented as a red and green scale to show the threshold of RFP and GFP signals marked by threshold function of ImageJ. Scale bars represent 100 μm. **(B, C)** Quantification of GFP and RFP fluorescence in male fly brain and VNC that expressing *GAL4^SIFa.PT^* together with *UAS-myrGFP* and *UAS-trans-Tango* (two-tailed unpaired *t* test). In all plots and statistical tests. Data are presented as mean±s.e.m. ns=not significant ($p > 0.05$), *$p < 0.05$, **$p < 0.01$, ***$p < 0.001$, ****$p < 0.0001$. Sample sizes (*n*) are indicated in the figure panels. **(D)** Male flies expressing *SIFaR^24F06^-GAL4* drivers together with *UAS-myrGFP* were imaged live under a fluorescent microscope with anti-GFP (yellow), and nc82 (blue) antibodies. Areas outlined by boxes are enlarged in the bottom panel. The bottom panels are presented as a yellow scale to show the threshold of GFP signals marked by threshold function of ImageJ. Scale bars represent 100 μm in brain and CB, 10 μm in OL. **(E)** Comparative examination of AG region SIFa *trans*-Tango fluorescence signals and SIFaR expression. **(F)** Schematic illustration of SIFa neurons projecting to PB, AL, AI, PRW, AG, etc. to regulate different behaviors. **(G–J)** MD assays for *GAL4*-mediated expression

of *UAS-TNT* (G), *UAS-KCNJ2-eGFP* (H), *UAS-NachBac-eGFP* (I), *and UAS-OrkΔC* using *empty-GAL4* driver (two-tailed unpaired *t* test). In all plots and statistical tests. Data are presented as mean ± s.e.m. ns = not significant ($p > 0.05$), *$p < 0.05$, **$p < 0.01$, ***$p < 0.001$, ****$p < 0.0001$. Sample sizes (n) are indicated in the figure panels. Underlying data for all graphs can be found in file S1 Data.
(TIF)

**S1 File. MD assay summary of NPs and NP-Rs-RNAi screening with elavc155 driver, related to** Fig 1. (A–H′) MD assays for GAL4-mediated knockdown of AKH, AkhR, AstA, AstA-R2, MIP, AstC, AstC-R1, AstC-R2 (BL25940) (in the order of A–H) using the elavc155 driver (two-tailed unpaired t test). In all plots and statistical tests. Data are presented as mean ± s.e.m. ns = not significant ($p > 0.05$), *$p < 0.05$, **$p < 0.01$, ***$p < 0.001$, ****$p < 0.0001$. Sample sizes (*n*) are indicated in the figure panels.
(TIF)

**S2 File. MD assay summary of NPs and NP-Rs-RNAi screening with elavc155 driver, related to** Fig 1. (I–P′) MD assays for GAL4-mediated knockdown of AstC-R2 (BL36888), amn, CCHa1, CCHa1, CrZ, CrzR, CCAP-R, DH31-R (in the order of I–P) using the elavc155 driver (two-tailed unpaired t test). In all plots and statistical tests. Data are presented as mean ± s.e.m. ns = not significant ($p > 0.05$), *$p < 0.05$, **$p < 0.01$, ***$p < 0.001$, ****$p < 0.0001$. Sample sizes (*n*) are indicated in the figure panels.
(TIF)

**S3 File. MD assay summary of NPs and NP-Rs-RNAi screening with elavc155 driver, related to** Fig 1. (Q–X′) MD assays for GAL4 mediated knockdown of Dh44-R2, ETH-RNAi, ETHR, FMRFaR, ITP, ilp1, ilp2, ilp3 (in the order of Q-X) using the elavc155 driver (two-tailed unpaired t test). In all plots and statistical tests. Data are presented as mean ± s.e.m. ns = not significant ($p > 0.05$), *$p < 0.05$, **$p < 0.01$, ***$p < 0.001$, ****$p < 0.0001$. Sample sizes (*n*) are indicated in the figure panels.
(TIF)

**S4 File. MD assay summary of NPs and NP-Rs-RNAi screening with elavc155 driver, related to** Fig 1. (Y–F′) MD assays for GAL4-mediated knockdown of ilp4, ilp5, ilp6, ilp7, lp7, InR(31037), InR(31594), MsR2 (in the order of Y–f) using the elavc155 driver (two-tailed unpaired t test). In all plots and statistical tests. Data are presented as mean ± s.e.m. ns = not significant ($p > 0.05$), *$p < 0.05$, **$p < 0.01$, ***$p < 0.001$, ****$p < 0.0001$. Sample sizes (*n*) are indicated in the figure panels.
(TIF)

**S5 File. MD assay summary of NPs and NP-Rs-RNAi screening with elavc155 driver, related to** Fig 1. (G–H′) MD assays for GAL4 mediated knockdown of Nplp2, Proc-R, RYa-R, CCKLR-17D1, TkR86, 5-HT1A, 5-HT2A, 5-HT7 (in the order of g–n) using the elavc155 driver (two-tailed unpaired t test). In all plots and statistical tests. Data are presented as mean ± s.e.m. ns = not significant ($p > 0.05$), *$p < 0.05$, **$p < 0.01$, ***$p < 0.001$, ****$p < 0.0001$. Sample sizes (*n*) are indicated in the figure panels.
(TIF)

**S1 Table. Summary of neuropeptide receptors *RNAi* screening with *elav*c155 driver.**
(TIF)

**S1 Data. Underlying numerical data for graphs in** Figs 1A–1P; 2A–2M; 3B–3C; 4A–4R, 4U, 4V; 5B–5I, 5L–5P; 6B–6E, 6G–6J; S1D, S1G, S1H, S1J, S2A–S2F, S2I, S2K, S2L, S2N, S2Q; S3C, S3D, S3G, S3H, S3I, S3J, S3M, S3N; S4A–S4L; S5A–S5H; S6A–S6D, S6L, S6M; S7A–S7V and S8B, S8C, S8G–S8J.
(XLSX)

## Author contributions

**Conceptualization:** Woo Jae Kim.

**Data curation:** Yutong Song, Tianmu Zhang, Tae Hoon Ryu, Kyle Wong, Zekun Wu, Justine Schweizer, Woo Jae Kim.

**Formal analysis:** Yutong Song, Tianmu Zhang, Tae Hoon Ryu, Kyle Wong, Zekun Wu, Yanan Wei, Justine Schweizer, Khoi-Nguyen Ha Nguyen, Alex Kwan, Xiaoli Zhang, Woo Jae Kim.

**Funding acquisition:** Woo Jae Kim.

**Investigation:** Woo Jae Kim.

**Methodology:** Tae Hoon Ryu, Woo Jae Kim.

**Project administration:** Woo Jae Kim.

**Resources:** Woo Jae Kim.

**Supervision:** Woo Jae Kim.

**Validation:** Yutong Song, Tianmu Zhang, Woo Jae Kim.

**Visualization:** Zekun Wu, Woo Jae Kim.

**Writing – original draft:** Woo Jae Kim.

**Writing – review & editing:** Yutong Song, Kweon Yu, Woo Jae Kim.

## Acknowledgments

We thank Dr. Jan A. Veenstra (University of Bordeaux) for sharing SIFa$^{PT}$-GAL4 driver, Drs. Yuh Nung Jan and Lily Yeh Jan (UCSF, USA) for helpful comments and support on this paper. We are also very appreciative to the colleagues who supplied us with several fly strains: Dr. Wei Zhang (Tsinghua University), Fang Guo (Zhejiang University), and Dr. Yufeng Pan (Southeast University) and Drs. Young-Joon Kim and Sung-Eun Yoon (Korea Drosophila Resource Center, KDRC). The authors would like to express their gratitude to NVIDIA Academic Hardware Grant Program for providing GPU for behavioral analysis. This research was supported a University of Ottawa Startup grant 602496 to WJK, Startup funds from HIT Center for Life Science to WJK, a University of Ottawa Interdisciplinary Research Group Funding Opportunity (IRGFO stream 1 and 2) grants 148101 and 148747 to WJK, a Natural Sciences and Engineering Research Council of Canada (NSERC) Discovery grant (reference: 211406) to WJK, a University of Ottawa Brain and Mind Research Institute/ Center for Neural Dynamics Open call project grant 150950 to WJK, a Mitacs Globalink Research Internship Program grant 17268 to WJK. This research was also supported by the Brain Pool Program of the National Research Foundation in Korea grant ZYM5041911 to WJK, Burroughs Welcome Fund Collaborative Research Travel Grants (reference: 1017486) to WJK and a NVIDIA Academic Hardware Grant Program to WJK. The funders had no role in study design, data collection and analysis, decision to publish, or preparation of the manuscript. SGL received salary from the 'University of Ottawa Startup grant to WJK' and HM from the 'Startup funds from HIT Center for Life Science to WJK'.

Declaration of generative AI and AI-assisted technologies in the writing processDuring the creation of this work, the author(s) utilized 'Perplexity' to rephrase English sentences, verify English grammar, and detect plagiarism, as none of the authors of this paper are native English speakers. After using this tool/service, the author(s) reviewed and edited the content as needed and take(s) full responsibility for the content of the publication.

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
