## [Editor Report · Decision Letter 0]

18 Dec 2024

Dear Dr Kim, 

Thank you for submitting your manuscript entitled "Peptidergic neurons with extensive branching orchestrate the internal states and energy balance of male Drosophila melanogaster", together with the related manuscript, "Long-range neuropeptide relay as a mechanism for context-dependent interval timing behaviors" for consideration as Research Articles by PLOS Biology. Please accept my apologies for the delay in sending you an initial decision. Given that both of these studies were previously reviewed at Review Commons, and that both have been revised, the initial assessment took a bit longer than normal, as there was a lot of information to consider. 

I have now had a chance to discuss both of these papers, the reviews from Review Commons, and your response to reviewers, with one of our Academic Editors who has relevant expertise, and I am writing to let you know that we would like to send both of your submissions back to the original reviewers for another look.

However, before we can send your manuscript to reviewers, we need you to complete your submissions by providing the metadata that is required for full assessment. To this end, please login to Editorial Manager where you will find the paper in the 'Submissions Needing Revisions' folder on your homepage. Please click 'Revise Submission' from the Action Links and complete all additional questions in the submission questionnaire.

Once your full submission is complete, your paper will undergo a series of checks in preparation for peer review. After your manuscript has passed the checks it will be sent out for review. To provide the metadata for your submission, please Login to Editorial Manager (https://www.editorialmanager.com/pbiology) within two working days, i.e. by Dec 20 2024 11:59PM.

**Please note, that in a moment, I will send you another email inviting in the full submission for your related manuscript (PBIOLOGY-D-24-03578). You will need to complete the metadata for both of these studies. 

**And as a last note, the PLOS Biology office will be closed from Dec 23-Jan 3rd, and so there is a good chance that we will not be able to get the reviewers signed on until early in January. I am sorry in advance for the additional delay! 

Kind regards,

Luke

Lucas Smith, Ph.D.

Senior Editor

PLOS Biology

lsmith@plos.org

---

## [Decision Letter · Decision Letter 1]

7 Mar 2025

Dear Woo Jae, 

Thank you for your patience while your manuscript "Peptidergic neurons with extensive branching orchestrate the internal states and energy balance of male Drosophila melanogaster" was peer-reviewed at PLOS Biology. Please accept our apologies, again, for the delay in sending you a decision on this paper. As noted in my previous correspondence, we had sent this study back to the original reviewers, who reviewed your paper at Review Commons - but only Reviewer 3 was available to review this study again. As reviewers 1 and 2 were not available, we ended up recruiting a new reviewer to assess the study as a revision and that search added some time to the assessment process. It has also taken us a bit longer than normal to finalize our decision for your paper, as the decision has required a bit of back and forth discussion within the team. 

As you will see in the reviewer comments, which are appended below, the reviewers agree that the revision has addressed many of the previous issues with this study but they both have lingering concerns which we think should be addressed in another round of revision. Some of Reviewer 4’s comments relate to the presentation of the study – which we note was an issue raised in the earlier rounds of review. We encourage you to address this point to the extent possible by presenting the controls in the main figure (although if those were generated independently, this should be clearly indicated and the controls should not be added into the same figure with other samples), and adding clearer motivations, and addressing Reviewer 4’s ‘specific comments’. However, after discussion with the Academic Editor, it is our opinion that the manuscript is easy enough to follow and we would not require a major restructuring of the narrative, etc for publication. 

We do however, think that it will be important to provide additional, direct proof that SIFamide neurons are dopaminergic through anti-TH immunoreactivity as the reviewers suggest. And we think that the other major concerns from Reviewer 4 will need to be carefully attended to - although we think that reviewer 4’s comment request to ‘explain the fact that synaptic transmission inhibition of SIFa neurons using Shibire has no effect on mating duration but Vglut, ple or SIFa knockdown in SIFa neurons all have an effect’ can likely be addressed through discussion.

While we are willing to give you another chance to address these last issues, given our interest in your study, I should note that this is the last round of experimental revision that we will entertain and that we will be looking to see that the reviewer concerns are adequately addressed, including with new data, before we can consider your paper for publication at PLOS Biology. 

If it would be helpful, we are happy to discuss and provide input on a revision plan for your paper. 

Given the extent of revision needed, we cannot make a decision about publication until we have seen the revised manuscript and your response to the reviewers' comments. Your revised manuscript may be sent for further evaluation by all or a subset of the reviewers.

**IMPORTANT - SUBMITTING YOUR REVISION**

*Re-submission Checklist*

*Published Peer Review*

*PLOS Data Policy*

*Blot and Gel Data Policy*

Sincerely,

Luke

Lucas Smith, Ph.D.

Senior Editor

PLOS Biology

lsmith@plos.org

REVIEWS

Original Reviewer #3: The manuscript "Peptidergic neurons with extensive branching orchestrate the internal states and energy balance of male Drosophila melanogaster" by Song and colleagues is very interesting for a broad readership. The Authors have substantially addressed the comments raised by the reviewers. However, I still would have some comments before endorsing publication of the manuscript:

During the review process the authors were asked to show evidence that SIFamdergic neurons indeed express tyrosine hydroxylase. The authors show the expression of ple through Scope RNA seq data (Figure 1O, related to page 10, line 208) and indirectly by employing an expression toolkit for dopaminergic neurons. There are very reliable and well described antibodies against Tyrosine hydroxylase commercially available. In my opinion direct proof that SIFamide neurons are dopaminergic through anti-TH immunoreactivity (immunostar) would be important.

Further, in Figure 1L the authors the expression pattern of SIFa.PT-Gal4. The images on the left - both, the overview and the magnification the image shows some bizarre lines on the upper left part of the brain - the authors should improve the image quality.

In line 132 on page 6 - the authors write that SIFa "protein" level remained the same - seen the size of SIFamide it would be preferable if the authors stick to the term peptide instead of protein.

Lastly, it would be preferable that on page 18, line 360 the authors again refer to Martelli et al., as this reflects exactly the same experiment that was conducted by this group.

With exception of these very few points mentioned above I am very satisfied about the hard work the authors have invested to address the reviewers' comments.

New Reviewer #4 (who assessed the responses to reviewers 1 and 2): Title: Peptidergic neurons with extensive branching orchestrate the internal states and energy balance of male Drosophila melanogaster.

General comments: This manuscript by Song et al focuses on SIFa and its role in context-dependent mating duration. The authors show that feedback between sNPF and SIFa is important for this behaviour. Overall, I found it challenging to follow all the experiments presented in this manuscript. It was not always clear the motivation for performing some experiments. In other instances, unusually large amount of attention is given to experiments that seem superfluous e.g. knockdown of SIFa in glia or re-characterizing the morphology of SIFa neurons which don't add much to the overall message of the manuscript. While the authors have addressed some of the concerns raised by previous reviewers, I think that the overall message of the manuscript is hard to follow This is not made easy by the fact that the genetic controls are not presented with the experimental treatments in the same figures. The manuscript still needs extensive restructuring so that main message of the manuscript becomes clear. 

Major comments:

* Not all neurons in TH-C GAL4 are anti-TH positive. The authors can refer to the original paper by Xie et al where this line was characterized. As suggested by previous reviewers, stronger evidence is needed for the presence of dopamine in SIFa neurons. 

* How can the authors explain the fact that synaptic transmission inhibition of SIFa neurons using Shibire has no effect on mating duration but Vglut, ple or SIFa knockdown in SIFa neurons all have an effect. Surely, inhibition of the neurons is a more drastic manipulation than knockdown of each neurotransmitter. 

* FlyScope data are not convincing. Multiple markers are necessary to conclusively claim that the transcriptomes of interest belong to a particular cell type. Neuropeptide transcripts tend to be expressed in very high amounts and it can generate false positives (see: https://pubmed.ncbi.nlm.nih.gov/32314735/ ). 

* Figure 5A-E - the confocal images look oversaturated. I am not sure how one can accurately quantify the area of fluorescence based on these. 

Specific comments:

Figure 1A: is it not significant or is there a difference

Lines 114-116: I am not sure why this experiment is important since several previous studies have conclusively showed that SIFa is expressed in neurons and not glia. 

Lines 121-124: Since the authors don't show staining pattern for the GAL4 for the rest of the brain and VNC, it is difficult to conclude that the phenotype is due to SIFa knockdown in only the intended neuronal population. Figure S1B shows SIFa staining in several neurons in w1118 flies. It is certainly expressed in more than 4 neurons.

Figure S1C: SIFa staining looks unlike what has been reported previously. The authors also state that these neurons have extensive branching but I don't see any branching in their images. Given the quality of this imaging, it is difficult to make any conclusions on the SIFa protein levels. 

Lines 143-145: The representative images don't match the conclusion. It looks like the staining in males is more intense and there is increased arborization in female antennal lobes. So I am not entirely convinced by the conclusions of this analysis. 

Lines 147-153: The motivation for performing these experiments is not provided so the reader is left confused. 

The section on Architecture of SIFa-positive neurons in the central nervous system does not add to the manuscript in my opinion, especially since it is difficult to make sense of the evidence presented in Figure S3. 

Lines 204-206: Didn't the authors establish that SIFa is responsible for the LMD/SMD phenotype by knocking it down? What is the rationale that other NTs could be involved? 

"The Fly SCope single-cell RNA sequencing data indicates that SIFa neurons exhibit

the highest level of co-expression for VGlut and ple (Fig 2N-2O)." Based on FlySCope it is not possible to determine the most highly co-expressed gene along with SIFa. You can only compare a maximum of 3 genes at a time.

---

## [Editor Report · Decision Letter 2]

24 Jul 2025

Dear Woo Jae,

Thank you for your patience while we considered your revised manuscript "Peptidergic neurons with extensive branching orchestrate the internal states and energy balance of male Drosophila melanogaster." for publication as a Research Article at PLOS Biology. This revised version of your manuscript has been evaluated by the PLOS Biology editors and the Academic Editor. The Academic Editor is satisfied by the changes made in this revision and has commented that the revision does a nice job addressing the previous concerns. 

Based on our Academic Editor's assessment of your revision, we are likely to accept this manuscript for publication, however before we can do so, we need you to address a few last editorial points. These are detailed below. 

IMPORTANT: Please make sure to address the following data and other policy-related requests:

1) TITLE: We would like to suggest ta change to the title, to make it a bit more specific. If you agree, we suggest you change the title to: "SIFa peptidergic neurons orchestrate the internal states and energy balance of male Drosophila melanogaster'

2) DATA: As you are aware, PLOS has a Data Policy, which requires that all data be made available without restriction: http://journals.plos.org/plosbiology/s/data-availability. For more information, please also see this editorial: http://dx.doi.org/10.1371/journal.pbio.1001797

a. Supplementary files (e.g., excel). Please ensure that all data files are uploaded as 'Supporting Information' and are invariably referred to (in the manuscript, figure legends, and the Description field when uploading your files) using the following format verbatim: S1 Data, S2 Data, etc. Multiple panels of a single or even several figures can be included as multiple sheets in one excel file that is saved using exactly the following convention: S1_Data.xlsx (using an underscore).

b. Deposition in a publicly available repository. Please also provide the accession code or a reviewer link so that we may view your data before publication. 

>>Regardless of the method selected, please ensure that you provide the individual numerical values that underlie the summary data displayed in the following figure panels as they are essential for readers to assess your analysis and to reproduce it:

Fig1A-P; Fig 2A-M; Fig 3B-C; Fig 4A-R,U-V; Fig 5B-I,L-P; Fig 6B-E,G-J; 

Fig S1D,G-H,J, Fig S2A-F,I,K,L,N,Q; Fig S3C-J,M-N; Fig S4A-L; FigS5 A-H; Fig S6A-D,L-M; Fig S7A-V; Fig S8B-C,G-J;

>>Please also ensure that figure legends in your manuscript include information on where the underlying data can be found, and ensure your supplemental data file/s has a legend.

>>Please ensure that your Data Statement in the submission system accurately describes where your data can be found.

3) CODE: Per journal policy, if you have generated any custom code during the course of this investigation, please make it available without restrictions. Please ensure that the code is sufficiently well documented and reusable, and that your Data Statement in the Editorial Manager submission system accurately describes where your code can be found. 

We expect to receive your revised manuscript within two weeks. 

*Published Peer Review History*

*Press*

Sincerely,

Luke

Lucas Smith, Ph.D.

Senior Editor

lsmith@plos.org

PLOS Biology

---

## [Editor Report · Decision Letter 3]

1 Aug 2025

Dear Woo Jae,

Thank you for the submission of your revised Research Article "SIFa peptidergic neurons orchestrate the internal states and energy balance of male Drosophila melanogaster" for publication in PLOS Biology and thank you for addressing our last editorial requests in this revision. On behalf of my colleagues and the Academic Editor, Paul J Shaw, I am pleased to say that we can in principle accept your manuscript for publication, provided you address any remaining formatting and reporting issues. These will be detailed in an email you should receive within 2-3 business days from our colleagues in the journal operations team; no action is required from you until then. Please note that we will not be able to formally accept your manuscript and schedule it for publication until you have completed any requested changes.

PRESS

Sincerely, 

Lucas Smith, Ph.D.

Senior Editor

PLOS Biology

lsmith@plos.org